# Preserve and Sculpt: Manifold-Aligned Fine-tuning of Vision-Language Models for Few-Shot Learning

**Dexia Chen[1], Qianjie Zhu[3], Weibing Li[1], Yue Yu[2], Tong Zhang[2],[*] Ruixuan Wang[1,2][*]**
[1]Sun Yat-sen University, [2]Peng Cheng Laboratory, [3]Guangxi University

## Abstract

Pretrained vision-language models (VLMs), such as CLIP, have shown remarkable potential in few-shot image classification and led to numerous effective transfer learning strategies. These methods leverage the pretrained knowledge of VLMs to enable effective domain adaptation while mitigating overfitting through parameter-efficient tuning or instance-based consistency constraints. However, such regularizations often neglect the geometric structure of data distribution, which may lead to distortion of the overall semantic representation. To overcome this limitation, we propose a novel fine-tuning method, **M**anifold-**P**reserving and **S**culpting Tuning (**MPS-Tuning**). Regarding the data distribution in feature space as a semantic manifold, MPS-Tuning explicitly constrains the intrinsic geometry of this manifold while further sculpting it to enhance class separability. Specifically, MPS-Tuning preserves both macroscopic and microscopic topological structures of the original manifold by aligning Gram matrices of features before and after fine-tuning. Theoretically, this constraint is shown to approximate an upper bound of the Gromov-Wasserstein distance. Furthermore, features from the image and text modalities are paired, and pairwise similarities are optimized to enhance the manifold's class discriminability. Extensive experiments demonstrate that MPS-Tuning significantly improves model performance while effectively preserving the structure of the semantic manifold. [1]

## 1 Introduction

Vision-language models (VLMs), exemplified by CLIP (Radford et al., 2021), have made significant progress by training on massive image-text pairs using contrastive learning. These models create joint embedding spaces where images and texts with similar meanings are well aligned. For instance, the visual representation of a "cat" becomes positioned near the textual representation of "feline" but far from semantically distant concepts like "truck". This intuitive structure of the embedding space directly contributes to the models' strong generalization across diverse tasks.

However, preserving this intricate semantic structure presents significant challenges during task adaptation, especially in few-shot learning scenarios. Standard fine-tuning approaches tend to semantic structure collapse , where limited training samples cause catastrophic forgetting of pretrained representations, ultimately manifesting as severe degradation in generalization performance.

To address these challenges, two main paradigms of approaches have been proposed (Fig. 1). The first paradigm encompasses parameter-efficient fine-tuning (PEFT) methods, including prompt-based techniques such as CoOp (Zhou et al., 2022b) and adapter-based frameworks like CLIP-Adapter (Gao et al., 2024a), which mitigate overfitting by constraining the number of trainable parameters. The second paradigm comprises consistency-driven approaches, such as Prompt-SRC (Khattak et al., 2023b), which enforce consistency between the features or logits of individual samples before and after fine-tuning. Despite the demonstrated efficacy of these approaches, they either rely on implicit regularization of few-parameter fine-tuning, which limits model flexibility, or

---

[*]Corresponding author. Email: wangruix5@mail.sysu.edu.cn, zhangt02@pcl.ac.cn

[1]The code is available at https://github.com/kaderxon/MPS-Tuning.

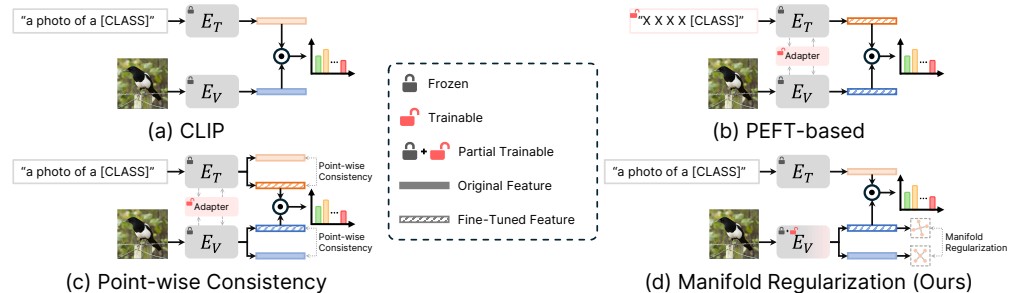

Figure 1: Comparison of regularization paradigms for VLMs. Previous fine-tuning methods (b, c) primarily constrain the number of tunable parameters or apply point-wise consistency constraints, potentially limiting the model's learning capacity while neglecting the complete semantic manifold structure. In contrast, our method explicitly preserves the semantic manifold structure, significantly enhancing both generalization and learning capabilities.

restrict variations in individual sample representations, which neglects the preservation of pretrained model's semantic structure.

In contrast to existing methods that treat image data as isolated points, we propose Manifold-Preserving and Sculpting Tuning (MPS-Tuning), which views data distribution in the feature spaces as continuous semantic manifold, and aims to enhance its discrimination for downstream tasks while maintaining the intrinsic manifold structure. To preserve the manifold structure, we constrain the Gromov-Wasserstein (GW) distance (Mémoli, 2011) between the semantic manifolds derived from the features distributions of the fine-tuned and original models during training. Since directly computing GW distance is NP-hard and impractical for optimization, we simplify this problem and theoretically prove that the $L_p$-norm of the difference between corresponding Gram matrices provides an upper bound approximation to the GW distance of order $p$. Based on this theoretical insight, we propose Manifold Alignment Regularization, which preserves global topological structure via batch-level Gram matrices and maintains local geometric structure through token-level Gram matrices. For manifold sculpting, we introduce Hierarchical Manifold Sculpting with a multimodal query-support matching task, where each query achieves higher similarity with same-category pairs and lower similarity with different-category pairs. This sculpting mechanism is extended from the model's output features to its intermediate layer features, further enhancing the discrimination of the manifold. Through manifold alignment and sculpting, robust adaptation of vision-language models is effectively achieved.

Our main contributions are summarized as follows:

- We propose a novel few-shot fine-tuning framework called MPS-Tuning, which enhances model performance while alleviating overfitting by explicitly aligning and sculpting the manifold geometry.
- We design a new regularization method, called Manifold Alignment Regularization, and establish its theoretical connection to the Gromov-Wasserstein distance for the first time, offering deeper insights into the preservation of manifold geometry.
- We introduce an optimization strategy called Hierarchical Manifold Sculpting to actively enhance the discriminability of the manifold and further improve model performance.
- We evaluated the method's performance on 11 datasets and conducted generalization evaluation on two datasets. Experimental results demonstrate that our method significantly outperforms current state-of-the-art approaches in few-shot image classification tasks.

## 2 RELATED WORK

### 2.1 VISION-LANGUAGE MODELS

In recent years, vision-language models (Radford et al., 2021; Sun et al., 2023; Xu et al., 2024; Gao et al., 2024b; Huang et al., 2024; Zhai et al., 2023; Tschannen et al., 2025; Pal et al., 2025)

pretrained via contrastive learning on large-scale image-text pairs have demonstrated strong zero-shot generalization capabilities by aligning correctly matched images and texts. This enables them to be directly applied to various downstream tasks. Typically, such models consist of an image encoder $E_V$ and a text encoder $E_T$. For a given classification task, text prompts (e.g., "a photo of a {class}") are first constructed for each class, and a set of normalized class-specific textual features $\{\mathbf{t}_1, ..., \mathbf{t}_K\}$ is extracted using the text encoder $E_T$, where $K$ denotes the number of classes. Subsequently, for an input image $\mathbf{x}$, its normalized visual feature representation $\mathbf{z}$ is obtained through the image encoder $E_V$. The probability of the image belonging to each class is then computed by applying the softmax function to the cosine similarities between $\mathbf{z}$ and each class text feature $\mathbf{t}_k$, i.e.,

$$P(y = c_k|\mathbf{x}) = \frac{\exp(\langle \mathbf{z}, \mathbf{t}_k \rangle/\tau)}{\sum_{j=1}^{K} \exp(\langle \mathbf{z}, \mathbf{t}_j \rangle/\tau)} , \tag{1}$$

where $\tau$ is a learnable temperature coefficient, and $\langle \cdot, \cdot \rangle$ represents the inner product between two vectors. The model's final prediction is given by the class with the highest probability, i.e.,

$$\hat{y} = \arg\max_{c_k \in \{c_1, ..., c_K\}} P(y = c_k|x) \tag{2}$$

## 2.2 EFFICIENT TRANSFER LEARNING

To adapt VLMs efficiently while mitigating overfitting, two main transfer learning paradigms have been explored (Yang et al., 2024; Zhu et al., 2023; Xie et al., 2024; khattak et al., 2025; Guo et al., 2023; Zhu et al., 2024). The first comprises PEFT-based methods that reduce computational cost through selective parameter updates, such as prompt-based approaches (e.g., CoOp (Zhou et al., 2022b), CoCoOp (Zhou et al., 2022a)) and adapter-based designs (e.g., CLIP-Adapter (Gao et al., 2024a), Tip-Adapter (Zhang et al., 2022)). The second paradigm emphasizes consistency regularization, exemplified by PromptSRC (Khattak et al., 2023b), which applies feature-level L1 loss and logit-level KL divergence (Kullback & Leibler, 1951) to retain pre-trained knowledge. While these methods demonstrate impressive performance on few-shot tasks, their flexibility is constrained by dependencies on PEFT strategies and point-based constraints. In contrast, our manifold alignment regularization enhances adaptability while better preserving pre-trained knowledge.

## 2.3 GROMOV–WASSERSTEIN DISTANCE

The Gromov–Wasserstein (GW) distance (Mémoli, 2011) compares the intrinsic geometric structures of two metric measure spaces by matching their internal pairwise distance relations rather than individual points. Consider two discrete metric spaces represented by their pairwise distance matrices $\mathbf{D}_{\mathcal{X}} \in \mathbb{R}^{n \times n}$ and $\mathbf{D}_{\mathcal{Y}} \in \mathbb{R}^{m \times m}$. Let $\mu \in \Delta_n$ and $\nu \in \Delta_m$ be probability vectors supported on these spaces, where $\Delta_n = \{u \in \mathbb{R}_+^n \mid \sum_{i=1}^{n} u_i = 1\}$, then the set of couplings between $\mu$ and $\nu$ is

$$\Pi(\mu, \nu) = \{\pi \in \mathbb{R}_+^{n \times m} \mid \pi \mathbf{1}_m = \mu, \ \pi^\top \mathbf{1}_n = \nu\}, \tag{3}$$

where $\pi_{ik}$ represents the joint probability mass assigned to the pair $(x_i, y_k)$, and $\mathbf{1}_m \in \mathbb{R}^m, \mathbf{1}_n \in \mathbb{R}^n$ are vectors of ones.

For $p \geq 1$, the discrete GW distance is defined as

$$GW_p(\mu, \nu) = \left( \min_{\pi \in \Pi(\mu, \nu)} \sum_{i,j=1}^{n} \sum_{k,l=1}^{m} |(\mathbf{D}_{\mathcal{X}})_{ij} - (\mathbf{D}_{\mathcal{Y}})_{kl}|^p \pi_{ik} \pi_{jl} \right)^{1/p} . \tag{4}$$

The GW objective seeks an optimal coupling $\pi$ that minimizes the expected p-power discrepancy between pairwise distances in the two spaces. By operating on the relational information encoded in the distance matrices, rather than the coordinates of individual points, this objective naturally becomes invariant to isometric relabelings. This invariance is precisely what enables GW distance to robustly capture and compare intrinsic geometric structures. However, finding $\pi$ is a significant bottleneck, as it requires solving a nonconvex quadratic program reducible to the NP-hard quadratic assignment problem, making it intractable for large-scale applications. To address this, we adopt a fixed coupling scheme, which provides a tractable upper bound on the GW distance and transforms it into a computationally feasible regularization. To the best of our knowledge, this work represents the first application of GW distance theory to VLM fine-tuning, offering a principled approach to preserving geometric knowledge in pretrained models. More details are in Sec. B.

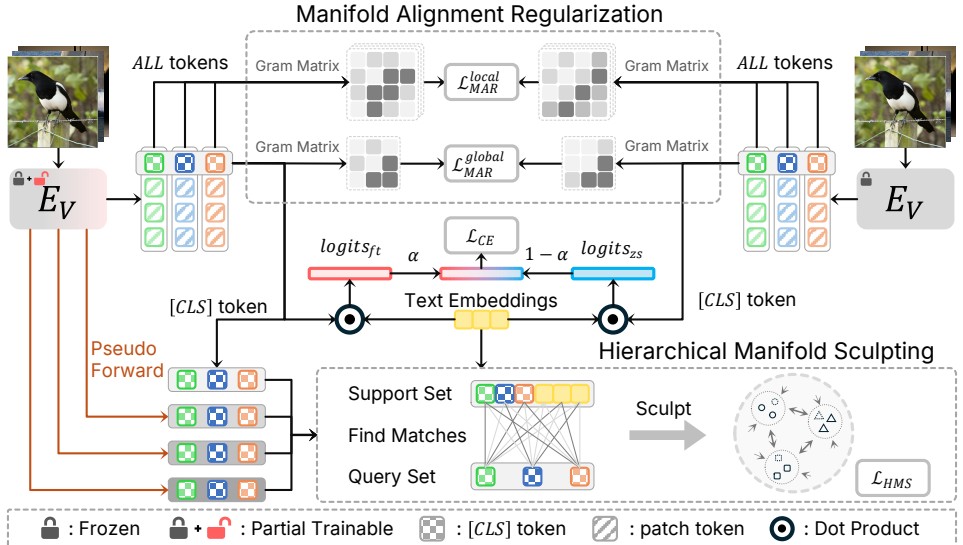

Figure 2: Overview of the MPS-Tuning, which integrates Manifold Alignment Regularization and Hierarchical Manifold Sculpting. Manifold Alignment Regularization prevents knowledge degradation by aligning Gram matrices across fine-tuned and original CLIPs at both global and local scales. Hierarchical Manifold Sculpting enhances local manifold adaptability via query-support matching, tailoring representations to downstream tasks. Through Pseudo Forward, this sculpting process extends to intermediate layers , ensuring effective manifold refinement. $E_V$ denotes the visual encoder.

## 3 METHOD

To facilitate effective downstream adaptation without disrupting the inherent structure of the pretrained representation manifold, we propose a novel approach termed Manifold-Preserving and Sculpting Tuning (MPS-Tuning). This method employs Manifold Alignment Regularization (MAR) to prevent drastic alterations in the semantic structure of feature manifold and incorporates Hierarchical Manifold Sculpting (HMS) to progressively refine local manifold structures. Specifically, MAR aligns the Gram matrices of the fine-tuned and the original models at both the batch and token levels, thereby maintaining consistency in semantic geometry and mitigating overfitting risks. In parallel, HMS refines local manifold structures by performing a multimodal query-support matching task between image and text representations, optimizing similarity at both intermediate and output feature levels. This results in more compact intra-class clusters and better separated interclass distributions. By jointly applying MAR and HMS, MPS-Tuning achieves robust and efficient adaptation to new tasks, maintaining the valuable structural knowledge of the pre-trained model and demonstrating strong performance in few-shot learning scenarios.

### 3.1 MANIFOLD ALIGNMENT REGULARIZATION

The feature distribution learned by a pre-trained model can be regarded as a well-structured semantic manifold, whose geometric structure encodes rich prior knowledge. Preserving this geometric structure allows for the retention of more comprehensive pre-trained knowledge. To this end, we propose Manifold Alignment Regularization (MAR), which enforces alignment between the geometric structures of feature manifolds before and after fine-tuning, thereby enhancing model performance.

As a metric designed to quantify the similarity between different metric spaces, the GW distance serves as a powerful tool for evaluating changes in the structure of feature manifolds induced by model fine-tuning. Our MAR provides an efficient upper-bound approximation to the GW distance. To formally justify this approximation, we present the following theorem:

**Theorem 1** *The alignment of the Gram matrices under the $L_p$-norm serves as an approximate upper bound of the $p$-order Gromov-Wasserstein distance.*

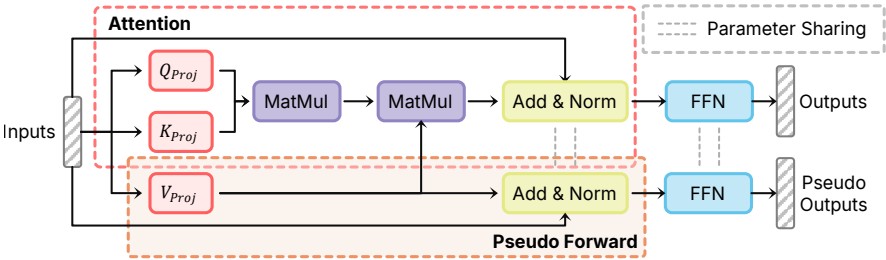

Figure 3: Pseudo-Forward projection bypass the attention allocation component in the model, and map intermediate layer features to the output feature space.

*Proof Outline.* We consider the feature spaces of the original CLIP model and the fine-tuned CLIP model as two metric spaces. By fixing a natural coupling (i.e., assuming a one-to-one correspondence between features of the same sample across the two models), the NP-hard computation of the GW distance is reduced to an efficient upper-bound approximation. Specifically, the approximate upper bound of the discrete $GW_p$ distance between the two metric spaces is given by the $L_p$-norm of the difference between their respective Gram matrices. Refer to Sec. B for more details.

Guided by this theory, alignment regularizations are introduced at two distinct levels.

**Global topological alignment** To preserve the global manifold structure, relational constraints among samples are enforced at the batch level. Given a mini-batch of $N$ samples, we extract normalized [CLS] token features from the pre-trained model as $\{\mathbf{z}_1, \ldots, \mathbf{z}_N\} \in \mathbb{R}^{N \times d}$ and from the fine-tuned model as $\{\mathbf{z}'_1, \ldots, \mathbf{z}'_N\} \in \mathbb{R}^{N \times d}$. The Gram matrices $\mathbf{S}, \mathbf{S}' \in \mathbb{R}^{N \times N}$ are computed via inner products $S_{ij} = \langle \mathbf{z}_i, \mathbf{z}_j \rangle$ and $S'_{ij} = \langle \mathbf{z}'_i, \mathbf{z}'_j \rangle$. The global alignment loss is defined as

$$\mathcal{L}_{\text{MAR}}^{\text{global}} = \frac{1}{N^2} \sum_{i=1}^{N} \sum_{j=1}^{N} |S_{ij} - S'_{ij}|_1 . \tag{5}$$

**Local geometric alignment** To retain the internal geometric structure within individual samples, regularization is separately performed on the interactions between the [CLS] token and the patch tokens, as well as on the internal relations among the patch tokens. For the $i$-th sample, we collect features before fine-tuning as $\{\mathbf{z}_{\text{cls}}^{(i)}, \mathbf{z}_1^{(i)}, \ldots, \mathbf{z}_M^{(i)}\} \in \mathbb{R}^{(M+1) \times d}$ and after fine-tuning as $\{\mathbf{z}'_{\text{cls}}^{(i)}, \mathbf{z}'_1^{(i)}, \ldots, \mathbf{z}'_M^{(i)}\} \in \mathbb{R}^{(M+1) \times d}$, where $M$ denotes the number of patch tokens. The intra-sample Gram matrices $\mathbf{S}_i^{\text{intra}}, \mathbf{S}'^{\text{intra}}_i \in \mathbb{R}^{(M+1) \times (M+1)}$ are computed using inner products among [CLS] and patch tokens. The local alignment loss is given by

$$\mathcal{L}_{\text{MAR}}^{\text{local}} = \frac{1}{N} \sum_{i=1}^{N} \left( \frac{1}{(M+1)^2} \sum_{k=0}^{M} \sum_{l=0}^{M} \left| S_{i,kl}^{\text{intra}} - S'^{\text{intra}}_{i,kl} \right|_1 \right) . \tag{6}$$

The final manifold alignment regularization term is the sum of the above two terms, i.e.,

$$\mathcal{L}_{\text{MAR}} = \mathcal{L}_{\text{MAR}}^{\text{global}} + \mathcal{L}_{\text{MAR}}^{\text{local}} . \tag{7}$$

### 3.2 HIERARCHICAL MANIFOLD SCULPTING

To facilitate the acquisition of task-specific knowledge, we propose a hierarchical optimization of the local feature manifold. This process is formulated as a query-support matching task, which aims to encourage high similarity for positive image-text or image-image pairs and discourage incorrect matches. Let $\mathcal{Q} = \{\mathbf{q}_1, \mathbf{q}_2, \ldots, \mathbf{q}_N\}$ be the set of normalized image features used as queries, and let $\mathcal{T} = \{\mathbf{t}_1, \mathbf{t}_2, \ldots, \mathbf{t}_K\}$ be the set of frozen text embeddings. The support set is then defined as the union $\mathcal{S} = \mathcal{Q} \cup \mathcal{T}$. Positive matches are defined based on category identity. Since limited batch sizes may lead to missing visual positives, data augmentation is applied to generate two augmented views per image, enriching the image pool. The task is optimized via a sculpting loss, which applies contrastive learning between each query and its positive matches:

$$\mathcal{L}_{sculpt}^{query}(\mathbf{q}, \mathcal{S}) = -\frac{1}{|\mathcal{P}_{\mathbf{q}}|} \sum_{\mathbf{s} \in \mathcal{P}_{\mathbf{q}}} \log \frac{\exp(\langle \mathbf{q}, \mathbf{s} \rangle)/\tau'}{\sum_{\mathbf{s}' \in \mathcal{S} \setminus \mathbf{q}} \exp(\langle \mathbf{q}, \mathbf{s}' \rangle)/\tau'} , \tag{8}$$

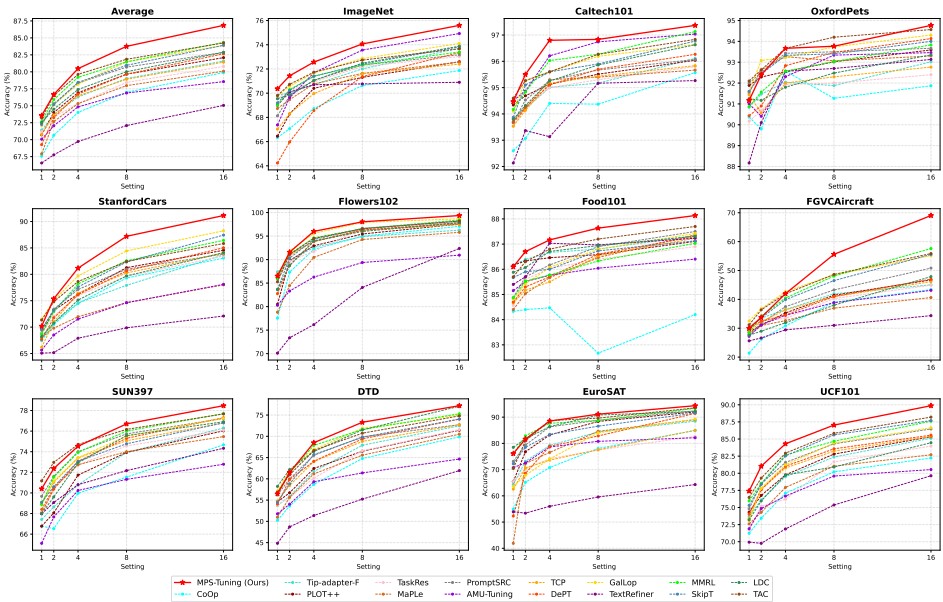

Figure 4: Performance comparison on 11 benchmark datasets.

where $\tau'$ is a temperature factor and $\mathcal{P}_q$ represents the set of samples from $\mathcal{S}$ that are positively paired with the query $\mathbf{q}$. The final objective over the batch is obtained by averaging this loss across all queries:

$$\mathcal{L}_{sculpt}(\mathcal{Q}, \mathcal{S}) = \mathbb{E}_{\mathbf{q} \in \mathcal{Q}}[\mathcal{L}_{sculpt}^{query}(\mathbf{q}, \mathcal{S})], \tag{9}$$

Furthermore, the manifold refinement is extended to intermediate transformer blocks to further sculpt the manifold. However, intermediate features $\mathbf{z}'^{(l)}$ are not compatible with text embeddings. To address this issue, we implement a pseudo-forward projection (Fig. 3), skipping attention modules while retaining essential transformations in model , i.e.,

$$\hat{\mathbf{z}}'^{(l)} = \text{FFN}^{(L)} \circ V_{\text{Proj}}^{(L)} \circ \cdots \circ \text{FFN}^{(l+1)} \circ V_{\text{Proj}}^{(l+1)}(\mathbf{z}'^{(l)}). \tag{10}$$

The overall HMS loss thus aggregates both final and intermediate layer alignment as follows,

$$\mathcal{L}_{\text{HMS}} = \mathcal{L}_{sculpt}(\hat{\mathcal{Q}}, \hat{\mathcal{S}}) + \sum_{l \in L_{\text{blocks}}} \mathcal{L}_{sculpt}(\mathcal{Q}^{(l)}, \mathcal{S}^{(l)}) \tag{11}$$

where $\hat{\mathcal{Q}}, \hat{\mathcal{S}}$ are the output query and support sets, $\mathcal{Q}^{(l)}, \mathcal{S}^{(l)}$ are their counterparts at layer $l$ after pseudo forward projection, and $L_{blocks}$ denotes the layer scope of HMS.

### 3.3 TRAINING AND INFERENCE

Leveraging the MPS-Tuning's strong knowledge retention capability, we can fine-tune partial model weights directly without causing overfitting, thereby substantially enhancing the model's learning capacity. See Sec. D for details.

During training and inference, the final logits are a weighted sum of fine-tuned and original outputs:

$$\text{logits} = \alpha \cdot \text{logits}_{\text{ft}} + (1 - \alpha) \cdot \text{logits}_{\text{zs}}. \tag{12}$$

The overall training loss incorporates the cross-entropy term alongside two regularization terms:

$$\mathcal{L} = \mathcal{L}_{\text{CE}} + \lambda_1 \mathcal{L}_{\text{MAR}} + \lambda_2 \mathcal{L}_{\text{HMS}} \tag{13}$$

where $\lambda_1$ and $\lambda_2$ are hyperparameters that balance the contributions of the regularization terms.

## 4 EXPERIMENTS

### 4.1 EXPERIMENTAL SETTINGS

**Datasets** Following previous work (Zhou et al., 2022b; Yu et al., 2023), we evaluated our method on 11 datasets, including ImageNet (Deng et al., 2009), Caltech101 (Fei-Fei et al., 2007), Food101 (Bossard et al., 2014), DTD (Cimpoi et al., 2014), EuroSAT (Helber et al., 2019), FGV-CAircraft (Maji et al., 2013), Flowers102 (Nilsback & Zisserman, 2008), OxfordPets (Parkhi et al., 2012), StanfordCars (Krause et al., 2013), SUN397 (Xiao et al., 2010), and UCF101 (Soomro et al., 2012). Additionally, domain generalization capabilities were further assessed using the ImageNet-Sketch (Wang et al., 2019) and ImageNet-V2 (Recht et al., 2019) datasets.

**Implementation** Following previous work (Zhou et al., 2022b; Yu et al., 2023), the model was trained on K-shot settings (K = 1, 2, 4, 8, 16) and evaluated on the full test set using CLIP with ViT-B/16 (Dosovitskiy et al., 2021) and predefined text templates. Optimization was performed using SGD with cosine learning rate decay over 50 epochs, where a warm-up strategy increased the learning rate linearly from 1e-5 to 0.002 during the first epoch. Data augmentation strategies consistent with those used in CoOp, including random cropping and random flipping, were applied. For hyperparameter configuration, the weights for MAR ($\lambda_1$), HMS ($\lambda_2$) and logits weight ($\alpha$) were set to 0.5, 0.1 and 0.3, respectively, with HMS applied to the last two layers. All results were averaged over three random seeds.

**Baselines** To demonstrate the superiority of our method, comprehensive comparisons were conducted against several SOTA methods, including CoOp (Zhou et al., 2022b), Tip-Adapter-F (Zhang et al., 2022), PLOT++ (Chen et al., 2023), MaPle (Khattak et al., 2023a), PromptSRC (Khattak et al., 2023b), AMU-Tuning (Tang et al., 2024), TCP (Yao et al., 2024), DePT (Zhang et al., 2024), GaLLop (Lafon et al., 2024), TextRefiner (Xie et al., 2024), MMRL (Guo & Gu, 2025), SkipT (Wu et al., 2025), LDC (Li et al., 2025), and TAC (Hao et al., 2025).

### 4.2 EFFICACY STUDY

**Classification Results** As shown in Fig. 4, our method achieves superior average performance across all shot settings compared to competing methods, with the performance gap widening as the number of training samples increases. Specifically, under the 1-shot, 4-shot, and 16-shot conditions, our approach improves accuracy by 0.88%, 1.27%, and 2.51%, respectively, over the strongest baseline. On natural image datasets such as ImageNet and SUN397, where the pre-trained CLIP model has already encoded rich visual knowledge, MAR facilitates the integration of downstream task learning while preserving this prior knowledge, leading to significant performance gains. For datasets with greater cross-domain challenges, such as StanfordCars, FGVCAircraft, and UCF101, the synergistic operation of HMS and MAR enables our method to effectively balance novel knowledge acquisition with pre-trained knowledge retention, yielding significant performance advantages.

**Domain Generalization Results** To validate the robustness of MPS-Tuning, models were trained on the ImageNet dataset and subsequently evaluated on both ImageNet-V2 and ImageNet-Sketch datasets. As demonstrated in Tab. 1, MPS-Tuning consistently outperforms all baseline methods across the three datasets, thereby confirming its superior domain generalization capabilities.

**Efficiency** The training and inference FPS of MPS-Tuning were evaluated on the SUN397 dataset on a single RTX 3090 GPU to assess its efficiency. As shown in Tab. 3, the method demonstrates comparable efficiency to existing approaches.

### 4.3 ABLATION STUDY

**Impact of Model Components** To establish the individual contribution of each proposed module, ablation studies were conducted on the ImageNet, StanfordCars, and SUN397 datasets under 16-shot settings. Tab. 2 illustrates that both modules contribute meaningful performance gains over the standard cross-entropy loss when applied separately, while their joint application achieves additional improvements.

**Impact of HMS depth** The application scope of HMS was further explored. As shown in Fig. 5, applying HMS to final layer and penultimate layer yields the best performance gain,

Table 1: Generalization results on ImageNet and its variants.

| Method | Source | Target | | Avg |
|---|---|---|---|---|
| | ImageNet | -Sketch | -V2 | |
| CLIP | 66.73 | 46.15 | 60.83 | 57.90 |
| Linear Probe CLIP | 65.85 | 34.77 | 56.26 | 52.29 |
| CoOp | 71.92 | 46.71 | 64.18 | 60.94 |
| PLOT++ | 72.48 | 47.13 | 65.07 | 61.56 |
| MaPLe | 72.56 | 49.20 | 64.10 | 61.95 |
| PromptSRC | 73.17 | 49.10 | 65.70 | 62.66 |
| TCP | 72.40 | 48.17 | 64.83 | 61.80 |
| DePT | 73.35 | 46.43 | 64.63 | 61.47 |
| AMU-Tuning | 74.93 | **50.37** | 65.42 | 63.57 |
| TextRefiner | 70.90 | 48.07 | 63.37 | 60.78 |
| MMRL | 72.03 | 49.17 | 64.47 | 61.89 |
| SkipT | 72.77 | 49.73 | 65.67 | 62.72 |
| LDC | 73.88 | 48.85 | 66.10 | 62.94 |
| TAC | 73.67 | 48.93 | 66.23 | 62.94 |
| MPS-Tuning (Ours) | **75.60** | 50.10 | **67.53** | **64.41** |

Table 2: Ablation study on different components. "Avg11" is the average over all 11 benchmark datasets.

| Components | | | Datasets | | | |
|---|---|---|---|---|---|---|
| $\mathcal{L}_{CE}$ | $\mathcal{L}_{MAR}$ | $\mathcal{L}_{HMS}$ | ImageNet | Cars | SUN397 | Avg11 |
| ✓ | | | 72.93 | 90.00 | 76.30 | 85.41 |
| ✓ | ✓ | | 75.30 | 90.80 | 78.07 | 86.44 |
| ✓ | | ✓ | 74.77 | 90.77 | 77.80 | 86.20 |
| ✓ | ✓ | ✓ | **75.60** | **91.13** | **78.47** | **86.85** |

Table 3: Efficiency comparison on SUN397.

| Method | Training FPS | Inference FPS |
|---|---|---|
| CLIP | - | 617 |
| CoCoOp | 4.20 | 13.0 |
| TCP | 120.9 | 617 |
| TextRefiner | 85.30 | 553 |
| MPS-Tuning (Ours) | 95.65 | 535 |

Table 4: Ablation study on MAR. "Avg11" is the average over all 11 benchmark datasets.

| Method | 1-shot | | | 4-shot | | | 16-shot | | |
|---|---|---|---|---|---|---|---|---|---|
| | ImageNet | UCF101 | Avg11 | ImageNet | UCF101 | Avg11 | ImageNet | UCF101 | Avg11 |
| None | 69.33 | 76.73 | 72.35 | 71.57 | 83.63 | 79.60 | 74.77 | 89.23 | 86.20 |
| only Global | 70.03 | 77.37 | 73.42 | 72.23 | 83.90 | 80.18 | 75.17 | 89.53 | 86.57 |
| only Local | 69.90 | 76.73 | 72.82 | 72.43 | 84.27 | 80.21 | 75.57 | 89.60 | 86.67 |
| MAR (Global+Local) | **70.37** | **77.40** | **73.55** | **72.57** | **84.30** | **80.47** | **75.60** | **89.87** | **86.85** |

Table 5: Ablation study of different consistency constraints. "Avg11" is the average over all 11 benchmark datasets.

| Consistency Constraint | | 1-shot | | | 4-shot | | | 16-shot | | |
|---|---|---|---|---|---|---|---|---|---|---|
| | | ImageNet | UCF101 | Avg11 | ImageNet | UCF101 | Avg11 | ImageNet | UCF101 | Avg11 |
| Feature-based | $cos$ | 69.73 | 75.37 | 72.77 | 71.83 | 82.63 | 79.36 | 74.73 | 88.20 | 86.07 |
| | $\ell_1$ | 69.43 | 76.73 | 72.41 | 71.63 | 83.63 | 79.66 | 74.83 | 89.23 | 86.24 |
| | $\ell_2$ | 69.33 | 76.80 | 72.35 | 71.63 | 83.60 | 79.65 | 74.77 | 89.17 | 86.20 |
| Logits-based | $kl$ | 69.67 | 73.00 | 71.43 | 71.60 | 78.70 | 77.77 | 73.87 | 84.97 | 84.20 |
| | $\ell_1$ | 69.50 | 75.00 | 71.91 | 71.50 | 81.03 | 78.22 | 74.67 | 87.67 | 85.65 |
| | $\ell_2$ | 69.47 | 74.93 | 71.85 | 71.57 | 81.10 | 77.51 | 74.70 | 87.73 | 85.59 |
| Manifold-based | $\mathcal{L}_{MAR}$ | **70.37** | **77.40** | **73.55** | **72.57** | **84.30** | **80.47** | **75.60** | **89.87** | **86.85** |

while extending it to earlier layers results in degradation. This finding aligns with the semantic hierarchy in neural networks (Zeiler, 2014; Krizhevsky et al., 2012; Gandelsman et al., 2024): deeper layers encode class-specific features that benefit from HMS, whereas intermediate layers capture more generic representations. Applying HMS too early may induce premature specialization, leading to overfitting. Therefore, HMS is used in the last two layers to enhance high-level representation learning while maintaining generalization.

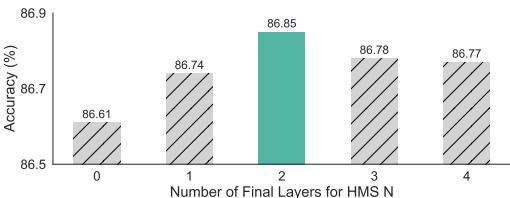

Figure 5: Ablation study on HMS depth. The x-axis $N$ means applying HMS to the last $N$ layers.

**Impact of Global and Local Alignment** To assess the roles of global topological and local geometric alignment, we perform ablation studies under 1-shot, 4-shot, and 16-shot settings. As shown in Tab. 4, both alignments consistently improve performance, but exhibit different preferences across sample sizes. Global alignment is more beneficial in low-shot scenarios, while local alignment becomes more effective as sample size increases. This is because global alignment offers essential

Table 6: Performance of distinct tuning methods equipped with vs. without MPS-Tuning. "PFT" indicates our default partial fine-tuning, and "FFT" stands for full fine-tuning.

| Method | Setting | ImageNet | Caltech101 | OxfordPets | StanfordCars | OxfordFlowers | Food101 | FGVCAircraft | SUN397 | DTD | EuroSAT | UCF101 | Avg |
|---|---|---|---|---|---|---|---|---|---|---|---|---|---|
| FFT | | 61.27 | 87.80 | 75.07 | 47.33 | 55.07 | 74.20 | 13.60 | 56.93 | 33.30 | 41.40 | 56.03 | 54.73 |
| FFT+MPS-Tuning | | 66.67 | 90.97 | 84.23 | 59.17 | 71.27 | 82.87 | 19.07 | 64.13 | 36.50 | 43.47 | 61.77 | 61.83 |
| LoRA | 1-shot | 68.37 | 94.33 | 91.60 | 66.27 | 81.97 | 85.83 | 28.97 | 68.00 | 50.90 | **81.37** | 74.00 | 71.96 |
| LoRA+MPS-Tuning | | 70.10 | 94.30 | **91.83** | 69.20 | 81.23 | **86.27** | 29.73 | 69.03 | 50.07 | 78.00 | 74.47 | 72.20 |
| PFT | | 68.80 | 94.00 | 90.13 | 68.03 | 86.50 | 84.93 | 28.17 | 69.40 | **57.53** | 74.47 | 76.63 | 72.60 |
| PFT+MPS-Tuning | | 70.37 | 94.47 | 91.17 | 70.17 | 86.50 | 86.10 | 29.97 | 70.40 | 56.47 | 76.10 | 77.40 | 73.55 |
| FFT | | 61.70 | 88.20 | 72.80 | 49.40 | 66.20 | 76.30 | 15.77 | 59.00 | 35.97 | 51.37 | 56.13 | 57.53 |
| FFT+MPS-Tuning | | 66.47 | 92.83 | 84.93 | 60.87 | 80.30 | 83.83 | 20.77 | 64.13 | 43.87 | 54.30 | 64.93 | 65.20 |
| LoRA | 4-shot | 69.27 | 95.93 | 93.10 | 74.50 | 93.00 | 86.20 | 38.90 | 71.53 | 62.30 | **89.50** | 80.60 | 77.71 |
| LoRA+MPS-Tuning | | 71.57 | 96.27 | 93.23 | 77.47 | 93.00 | 87.10 | 41.37 | 72.83 | 62.40 | 88.07 | 81.93 | 78.66 |
| PFT | | 70.43 | 96.33 | 92.33 | 78.30 | 95.53 | 85.73 | 37.10 | 73.10 | 68.10 | 87.83 | 83.27 | 78.92 |
| PFT+MPS-Tuning | | 72.57 | **96.80** | **93.67** | 81.17 | **96.00** | 87.17 | 41.97 | 74.53 | 68.50 | 88.50 | 84.30 | 80.47 |
| FFT | | 63.67 | 90.23 | 78.60 | 51.33 | 78.53 | 79.23 | 18.63 | 62.47 | 47.73 | 65.43 | 60.23 | 63.28 |
| FFT+MPS-Tuning | | 68.83 | 93.07 | 85.00 | 63.93 | 85.47 | 82.90 | 22.67 | 64.80 | 55.10 | 69.00 | 68.03 | 68.98 |
| LoRA | 16-shot | 71.77 | 97.03 | 93.93 | 87.30 | 98.43 | 86.70 | 62.43 | 75.27 | 71.70 | **94.40** | 86.90 | 84.17 |
| LoRA+MPS-Tuning | | 73.90 | 97.17 | 94.43 | 88.50 | 98.43 | 87.87 | 64.20 | 76.87 | 73.37 | 93.90 | 87.80 | 85.13 |
| PFT | | 72.93 | 97.07 | 93.53 | 90.00 | 99.03 | 86.20 | 66.33 | 76.30 | 75.87 | 93.77 | 88.43 | 85.41 |
| PFT+MPS-Tuning | | **75.60** | **97.37** | **94.77** | **91.13** | **99.37** | **88.13** | **69.03** | **78.47** | **77.20** | 94.37 | **89.87** | **86.85** |

relational priors when the model lacks enough data to capture inter-class structure, whereas local alignment enhances robustness by preventing shortcut learning from incidental factors when more data is available. Combining both leads to the best performance, confirming their complementary nature at different structural levels.

**Comparison to other Consistency Constraints** The effectiveness of MAR was further validated through performance comparisons with standard point-wise consistency constraints applied at feature and logit levels. As shown in Tab. 5, under 1-shot, 4-shot, and 16-shot settings, our method outperformed the strongest baselines by 0.64%, 0.74%, and 0.77% on ImageNet, and by 0.78%, 0.81%, and 0.61% when averaged across all 11 datasets. These results confirm that manifold-based consistency regularization yields superior performance improvements compared to point-alignment-based consistency constraints.

**Applicability Study** We evaluate the versatility of MPS-Tuning by integrating it into both full-parameter fine-tuning and LoRA (Hu et al., 2021). As shown in Tab. 6, our method yields consistent performance improvements across these strategies, validating its broad applicability.

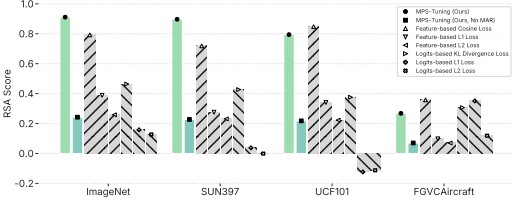

Figure 6: Representation similarity analysis of different consistency constraint mechanisms.

Figure 7: Class separability assessment of different component categories using Calinski-Harabasz score.

### 4.4 INTERPRETABILITY STUDY

**Quantitative Analysis of Manifold Preservation** To further validate MAR's manifold preservation capabilities, representational similarity analysis (Kriegeskorte et al., 2008) was applied to models before and after fine-tuning, enabling quantitative evaluation of semantic manifold structural variations. Fig. 6 shows that MAR achieves the best manifold structure preservation on natural image datasets like ImageNet and SUN397, where pretrained CLIP already contains sufficient knowledge. Furthermore, we find that MAR's preservation capability spontaneously diminishes on datasets with extensive novel task-relevant knowledge (UCF101, FGVCAircraft). This adaptive behavior results from MAR's relaxed Gram matrix alignment rather than rigid feature or logits constraints, enabling knowledge preservation when beneficial while allowing adaptation when necessary. Additionally, we find that cosine similarity and KL divergence impose stricter manifold structure preservation compared to other point-based consistency constraints through their holistic vector-level restrictions, whereas L1 and L2 losses independently constrain each channel's variation, potentially causing minimal per-channel changes but substantial overall vector modifications. However, their excessive strictness can limit learning (e.g., UCF101 and Avg11 results in Tab. 5), whereas MAR's Gram matrix consistency provides dynamic constraints that balance preservation with learning capability.

**Quantitative Evaluation of Class Separation** To assess feature clustering behavior, we apply the Calinski-Harabasz Index (Calinski et al., 1974) on test sets from ImageNet, SUN397, UCF101 and FGVCAircraft. As shown in Fig. 7, both HMS and MAR individually improve clustering performance compared to cross-entropy loss, with HMS performing particularly well on cross-domain dataset UCF101. The combination of HMS and MAR yields similar scores to those of HMS alone, suggesting that our method effectively improves feature separability.

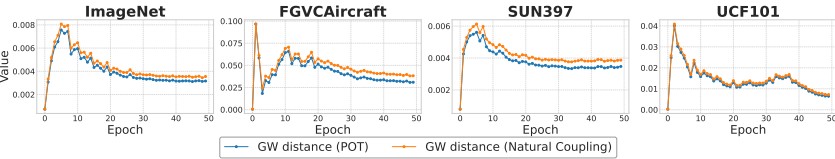

Figure 8: Comparison of GW distance: Natural coupling upper bound vs. POT numerical solution.

## 4.5 TIGHTNESS ANALYSIS

We approximate the upper bound of the $p$-th order GW distance via a fixed natural coupling. Its tightness is validated against exact GW distances calculated using POT (Flamary et al., 2021; 2024) on natural (ImageNet, SUN397) and cross-domain (FGVCAircraft, UCF101) datasets, with negligible divergence (Fig. 8) confirming its reliability and precision.

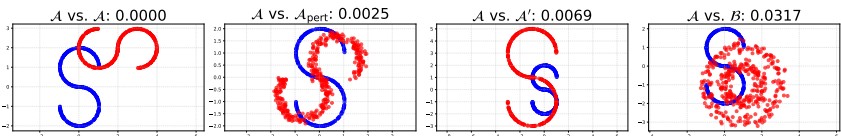

Figure 9: GW distances across manifold pairs.

## 5 DISCUSSION

### 5.1 GROMOV–WASSERSTEIN DISTANCE AND MANIFOLD ALIGNMENT

By minimizing intra-space distance discrepancies, the Gromov-Wasserstein distance aligns metric measure spaces while ensuring invariance to rigid transformations and sampling. We verify this using an S-shaped manifold $\mathcal{A}$ alongside its isometric variants $\mathcal{A}'$, noisy perturbations $\mathcal{A}_{pert}$, and topologically distinct rings $\mathcal{B}$. Results in Fig. 9 corroborate the isometric invariance and noise robustness of GW metrics, highlighting their capacity to distinguish topological discrepancies.

### 5.2 LIMITATION

While our approach demonstrates robust performance gains across the majority of scenarios, it falls short of SOTA results on specific 1-shot and 2-shot datasets. This could be attributed to the Gram matrix alignment strategy in MAR which struggles to capture the semantic manifold structure under extreme data scarcity. Future research could address this by incorporating unlabeled external data or employing generative models for data augmentation to enhance manifold structure preservation.

## 6 CONCLUSION

In this work, we propose MPS-Tuning, a manifold alignment-based fine-tuning framework that preserves the intrinsic structure of pre-trained models via Manifold Alignment Regularization, while enhancing the discriminability of semantic manifolds through Hierarchical Manifold Sculpting. We theoretically show that Manifold Alignment Regularization provides an approximate upper bound of the Gromov-Wasserstein distance, establishing its theoretical soundness. Extensive experiments demonstrate consistent improvements in few-shot VLM performance, positioning MPS-Tuning as a promising paradigm for advancing fine-tuning methodologies across diverse domains.

## ACKNOWLEDGMENTS

This work is supported in part by the National Natural Science Foundation of China (grant No. 62571559), the Major Key Project of PCL (grant No. PCL2025AS209), the Guangdong Excellent Youth Team Program (grant No. 2023B1515040025), the Guangdong Provincial Science and Technology Program (grant No. 2025A0505080012), and the Guangzhou Basic and Applied Basic Research Foundation (grant No. 2025A04J5280).

## ETHICS STATEMENT

This research adheres to the ICLR Code of Ethics. We utilize pre-trained vision-language models (e.g., CLIP) and are aware that such models may learn societal biases from their web-scale training data. Our work is a fundamental algorithmic study on improving few-shot learning performance, with experiments conducted solely on public academic datasets, involving no sensitive data or human subjects. We therefore consider the direct ethical risks of this research to be low. Nevertheless, we urge any researchers or developers applying this technology to real-world applications to be vigilant and proactively address potential issues of bias and fairness inherited from the foundation models.

## REPRODUCIBILITY STATEMENT

To support the reproducibility of this work, we provide comprehensive theoretical proofs and experimental details in the appendix. Detailed mathematical proofs for all theoretical claims in the paper can be found in Sec. B. All key information required to reproduce our experimental results, including detailed descriptions of the datasets (Sec. E) and complete hyperparameter configurations (Sec. C), is also provided. We plan to publicly release our source code upon publication.

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

## A  STATEMENT ON THE USE OF LARGE LANGUAGE MODELS (LLMS)

In the preparation of this paper, we utilized Large Language Models (LLMs), including ChatGPT, Gemini, Claude, Qwen, etc., as general-purpose assistive tools. Their primary use was for improving language expression, correcting grammatical errors, and polishing certain paragraphs to enhance the overall readability of the manuscript. We confirm that all core research ideas, experimental designs, result analyses, and final conclusions were conceived and executed by the human authors. The entire content of the paper was carefully reviewed and revised by the authors, who bear full responsibility for the final submission.

## B  DETAILED PROOF FOR THE UPPER BOUND ON GROMOV-WASSERSTEIN DISTANCE

**Theorem 2** *The alignment of Gram matrices under the $L_p$-norm serves as a tractable upper bound for the $p$-th order Gromov-Wasserstein distance ($GW_p$), when the underlying metric is the cosine distance.*

**Proof 1** *The objective is to formally demonstrate that the Manifold Alignment Regularization (MAR) strategy, which aligns Gram matrices, minimizes a computationally tractable upper bound of the $p$-th order Gromov-Wasserstein (GW) distance between the feature manifolds of the original pre-trained model and the fine-tuned model.*

**1. Definition of Metric Spaces.** *We consider two discrete metric probability spaces, $(\mathcal{X}, d_{\mathcal{X}}, \mu)$ and $(\mathcal{Y}, d_{\mathcal{Y}}, \nu)$.*

- ***Space $\mathcal{X}$:*** *Represents the feature space of the original, frozen model. For a mini-batch of $N$ samples, we extract a set of normalized features $Z = \{z_1, z_2, \ldots, z_N\}$, where $z_i \in \mathbb{R}^d$.*

- **Space $\mathcal{Y}$**: *Represents the feature space of the fine-tuned model. For the same mini-batch, the corresponding set of normalized features is $Z' = \{z'_1, z'_2, \ldots, z'_N\}$, where $z'_i \in \mathbb{R}^d$.*

- **Metric $d$**: *We employ the **Cosine Distance** as the metric. For a pair of normalized vectors $a$ and $b$, the distance is $d(a, b) = 1 - \langle a, b \rangle$. The intra-space distance matrices, $D_Z$ and $D_{Z'}$, are thus:*

$$(D_Z)_{ij} = d(z_i, z_j) = 1 - \langle z_i, z_j \rangle \tag{14}$$

$$(D_{Z'})_{ij} = d(z'_i, z'_j) = 1 - \langle z'_i, z'_j \rangle \tag{15}$$

- **Probability Distributions $\mu, \nu$**: *For a discrete batch of $N$ samples, we assume a uniform probability distribution, i.e., $\mu = \nu = \frac{1}{N} \sum_{i=1}^{N} \delta_i$, where $\delta_i$ is the Dirac measure.*

**2. Gromov-Wasserstein Distance and the Concept of Coupling.** *The discrete $p$-th order GW distance is formulated as:*

$$GW_p(\mu, \nu) = \left( \min_{\pi \in \Pi(\mu, \nu)} \sum_{i,j=1}^{N} \sum_{k,l=1}^{N} |(D_Z)_{ij} - (D_{Z'})_{kl}|^p \pi_{ik} \pi_{jl} \right)^{1/p} \tag{16}$$

*Here, $\pi \in \Pi(\mu, \nu)$ is a **coupling**, which is a joint probability distribution over the product space $\mathcal{X} \times \mathcal{Y}$. Intuitively, a coupling can be understood as a probabilistic transportation plan that describes how to map or associate the points from space $\mathcal{X}$ (with distribution $\mu$) to the points in space $\mathcal{Y}$ (with distribution $\nu$). The GW distance seeks the optimal coupling $\pi^*$ that minimizes the expected difference between pairwise distances in the two spaces. Finding this optimal plan involves solving a quadratic assignment problem, which is computationally intractable (NP-hard) for non-trivial cases.*

**3. Simplification via a Fixed Coupling.** *To derive a computationally feasible upper bound, we forgo the optimization over all possible couplings and instead select a single, fixed coupling. As outlined in the paper, we adopt a natural coupling, $\pi_{nat}$, which assumes a one-to-one correspondence between the features of the same sample before and after fine-tuning. This coupling is formally defined as:*

$$\pi_{nat,ik} = \begin{cases} 1/N & \text{if } i = k \\ 0 & \text{if } i \neq k \end{cases} \tag{17}$$

*By definition, the GW distance is the minimum over all couplings. Therefore, using our specific $\pi_{nat}$ provides an upper bound on the true GW distance.*

**4. Derivation of the Upper Bound.** *We substitute our fixed coupling $\pi_{nat}$ into the $p$-th power of the GW distance formula:*

$$GW_p(\mu, \nu)^p \leq \sum_{i,j=1}^{N} \sum_{k,l=1}^{N} |(D_Z)_{ij} - (D_{Z'})_{kl}|^p \cdot \pi_{nat,ik} \pi_{nat,jl} \tag{18}$$

*Due to the structure of $\pi_{nat}$, the term $\pi_{nat,ik} \pi_{nat,jl}$ is non-zero only when $i = k$ and $j = l$, where it evaluates to $(1/N)(1/N) = 1/N^2$. This simplifies the quadruple summation into a double summation:*

$$GW_p(\mu, \nu)^p \leq \frac{1}{N^2} \sum_{i=1}^{N} \sum_{j=1}^{N} |(D_Z)_{ij} - (D_{Z'})_{ij}|^p \tag{19}$$

**5. Connection to Manifold Alignment Regularization (MAR).** *Let $S$ and $S'$ denote the Gram matrices (cosine similarity matrices). The absolute difference in distances can be expressed in terms of the Gram matrices:*

$$\begin{aligned} |(D_Z)_{ij} - (D_{Z'})_{ij}| &= |(1 - \langle z_i, z_j \rangle) - (1 - \langle z'_i, z'_j \rangle)| \\ &= |S'_{ij} - S_{ij}| \end{aligned} \tag{20}$$

*Substituting this back into our inequality, we arrive at the final upper bound:*

$$GW_p(\mu, \nu)^p \leq \frac{1}{N^2} \sum_{i=1}^{N} \sum_{j=1}^{N} |S'_{ij} - S_{ij}|^p \tag{21}$$

*This expression shows that the $p$-th power of the GW distance is upper-bounded by the scaled $L_p$-norm of the difference between the Gram matrices of the two feature spaces.*

**Conclusion.** *We have formally shown that minimizing the $L_p$-norm of the difference between Gram matrices corresponds to minimizing a tractable upper bound on the $p$-th power of the Gromov-Wasserstein distance. The Manifold Alignment Regularization (MAR) loss presented in the paper, $\mathcal{L}_{MAR}$, is a specific instance of this principle using the $L_1$-norm ($p = 1$). This provides a strong theoretical justification for how aligning Gram matrices effectively preserves the geometric structure of the feature manifold during fine-tuning.*

Table 7: Default hyperparameter settings used for training MPS-Tuning.

| Hyperparameter | Value |
| --- | --- |
| Optimizer | SGD |
| Batch Size | 32 |
| Total Epochs | 50 |
| Peak Learning Rate | 0.002 |
| LR Scheduler | Cosine Decay |
| Logits Weight $\alpha$ | 0.3 |
| MAR Loss Weight $\lambda_1$ | 0.5 |
| HMS Loss Weight $\lambda_2$ | 0.1 |
| HMS Depth | 2 |

## C  HYPERPARAMETER SETTINGS

This section provides a detailed overview of the hyperparameter settings used in our experiments for MPS-Tuning. The primary hyperparameter values, which serve as the default for most datasets, are presented in Table 7. All experiments were conducted using the CLIP ViT-B/16 as the base model and were averaged over three different random seeds to ensure the reliability of our results.

While most parameters were kept consistent to ensure a fair evaluation, certain key hyperparameters were adjusted for specific datasets to optimize performance. Specifically, the weight for the Manifold Alignment Regularization loss ($\lambda_1$) was increased to 2.0 for datasets with natural images, namely ImageNet, OxfordPets, Food101, and SUN397, to enforce stronger preservation of the rich pre-trained manifold. For all other datasets, the default value of 0.5 was used. Similarly, the batch size was set to 64 for the large-scale ImageNet dataset to ensure stable gradient estimation, while a batch size of 32 was used for all other datasets.

Furthermore, we employed a dynamic temperature scheduling for the HMS loss ($\tau'$) using a cosine annealing strategy over the training epochs. For datasets including OxfordPets, Food101, DescribableTextures, EuroSAT, and UCF101, the temperature was annealed from an initial value of 0.5 down to 0.07. For the remaining datasets, a more conservative schedule from 0.1 to 0.05 was applied.

To mitigate the risk of overfitting caused by excessive category-specific supervision on intermediate layers, we incorporate a simple layer-wise decay scheme in HMS. Concretely, the final layer is assigned a weight of 1, and the weight of each preceding layer is defined recursively as half that of its subsequent layer:

$$w_L = 1, \quad w_l = \frac{1}{2} w_{l+1} \quad \text{for } l = L-1, L-2, \ldots, 1, \tag{22}$$

where $L$ denotes the total number of layers.

Regarding data augmentation, we followed the standard protocol used in CoOp. This includes 'RandomResizedCrop' with a scale range of (0.3, 1.0) and random horizontal flipping. No other complex augmentations were used.

## D  TRAINABLE MODULES

Benefiting from the powerful regularization capacity of MAR, direct fine-tuning of pre-trained models becomes feasible in few-shot scenarios. Specifically, a hierarchical fine-tuning strategy is employed for the visual encoder. The modules in the ViT/B-16 backbone are grouped based on semantic hierarchy (Gandelsman et al., 2024), with every four layers forming a group. The first group remains frozen during training to retain general representation learning. In the second group, each Transformer block is paired with a zero-initialized linear layer operating in parallel. Inputs are processed by both branches, and their outputs are summed to allow lightweight adjustments in intermediate representations. The third group is fully fine-tuned to facilitate adaptation to downstream tasks.

Table 8: Summary of 11 datasets for few-shot learning and 2 target datasets of domain generalization. The 7 selected templates (Zhang et al., 2022) for ImageNet series datasets are "itap of a [class].", "a bad photo of the [class].", "a origami [class].", "a photo of the large [class].", "a [class] in a video game.", "art of the [class]." and "a photo of the small [class]."

| Name | Number of Classes | Size (Train / Val / Test) | Description | Template |
|---|---|---|---|---|
| ImageNet (Deng et al., 2009) | 1000 | 1.28M / - /50000 | Recognition of generic objects | |
| ImageNet-V2 (Recht et al., 2019) | 1000 | - / - / 10000 | New test data for ImageNet | Ensemble of 7 selected templates |
| ImageNet-Sketch (Wang et al., 2019) | 1000 | - / - / 50889 | Sketch-style images of ImageNet classes | |
| Caltech101 (Fei-Fei et al., 2007) | 100 | 4128 / 1649 / 2465 | Recognition of generic objects | "a photo of a [class]." |
| OxfordPets (Parkhi et al., 2012) | 37 | 2944 / 736 / 3669 | Fine-grained classification of pets | "a photo of a [class], a type of pet." |
| StanfordCars (Krause et al., 2013) | 196 | 6509 / 1635 / 8041 | Fine-grained classification of cars | "a photo of a [class]." |
| Flowers102 (Nilsback & Zisserman, 2008) | 102 | 4093 / 1633 / 2463 | Fine-grained classification of flowers | "a photo of a [class], a type of flower." |
| Food101 (Bossard et al., 2014) | 101 | 50500 / 20200 / 30300 | Fine-grained classification of foods | "a photo of a [class], a type of food." |
| FGVCAircraft (Maji et al., 2013) | 100 | 3334 / 3333 / 3333 | Fine-grained classification of aircrafts | "a photo of a [class], a type of aircraft." |
| SUN397 (Xiao et al., 2010) | 397 | 15880 / 3970 / 19850 | Scene classification | "a photo of a [class]." |
| DTD (Cimpoi et al., 2014) | 47 | 2820 / 1128 / 1692 | Texture classification | "[class] texture." |
| EuroSAT (Helber et al., 2019) | 10 | 13500 / 5400 / 8100 | Land use & cover classification with satellite images | "a centered satellite photo of [class]." |
| UCF101 (Soomro et al., 2012) | 101 | 7639 / 1898 / 3783 | Action recognition | "a photo of a person doing [class]." |

## E  DATASETS

In the main text, our method was assessed on the widely adopted CLIP Benchmark, in alignment with previous work (Zhou et al., 2022b; Zhang et al., 2022; Yu et al., 2023). The benchmark comprises 11 diverse datasets, including ImageNet (Deng et al., 2009), Caltech101 (Fei-Fei et al., 2007), Oxford Pets (Parkhi et al., 2012), Stanford Cars (Krause et al., 2013), Flowers102 (Nilsback & Zisserman, 2008), Food101 (Bossard et al., 2014), FGVCAircraft (Maji et al., 2013), SUN397 (Xiao et al., 2010), DTD (Cimpoi et al., 2014), EuroSAT (Helber et al., 2019), and UCF101 (Soomro et al., 2012). These datasets span a broad range of image classification scenarios, encompassing general object recognition, fine-grained object recognition, scene recognition, texture recognition, and satellite imagery analysis, which allows for a thorough assessment of our model's generalization capabilities across various domains. To ensure consistency with previous work (Zhou et al., 2022b; Zhang et al., 2022; Yu et al., 2023), the "BACKGROUND Google" and "Faces easy" classes were excluded from the Caltech101 dataset. Additionally, robustness under domain shift was analyzed using two ImageNet variants: ImageNet-V2 (Recht et al., 2019), containing 200 overlapping classes, and ImageNet-Sketch (Wang et al., 2019), encompassing 1,000 classes identical to ImageNet. Consistent with earlier works, ImageNet was used as the source dataset, while the two variants served as target datasets. An overview of these datasets is presented in Tab. 8.

## F  NUMERICAL RESULTS

### F.1  CLASSIFICATION RESULTS

Comparative evaluations (Tab. 9) were conducted across 11 benchmark datasets against state-of-the-art methods, including CoOp (Zhou et al., 2022b), Tip-Adapter-F (Zhang et al., 2022), PLOT++ (Chen et al., 2023), MaPle (Khattak et al., 2023a), PromptSRC (Khattak et al., 2023b),

Table 9: Performance comparison on CLIP benchmark on ViT-B/16.

| Method | Setting | ImageNet | Caltech101 | OxfordPets | StanfordCars | Flowers102 | Food101 | FGVCAircraft | SUN397 | DTD | EuroSAT | UCF101 | Average |
|---|---|---|---|---|---|---|---|---|---|---|---|---|---|
| CLIP (Radford et al., 2021) | | 66.73 | 93.35 | 88.25 | 65.48 | 67.44 | 83.65 | 23.67 | 62.59 | 44.27 | 42.01 | 65.13 | 63.87 |
| CoOp (Zhou et al., 2022b) | | 66.33 | 92.60 | 90.37 | 67.43 | 77.53 | 84.33 | 21.37 | 66.77 | 50.23 | 54.93 | 71.23 | 67.56 |
| Tip-adapter-F (Zhang et al., 2022) | | 69.83 | 93.83 | 90.84 | 67.88 | 87.37 | 86.17 | 30.39 | 67.42 | 53.72 | 64.35 | 73.70 | 71.41 |
| PLOT++ (Chen et al., 2023) | | 66.45 | 94.34 | 91.89 | 68.81 | 80.48 | 86.16 | 28.60 | 66.77 | 54.57 | 65.41 | 74.31 | 70.71 |
| TaskRes (Yu et al., 2023) | | 69.57 | 93.53 | 90.17 | 68.83 | 85.77 | 84.57 | 31.30 | 68.13 | 53.80 | 65.43 | 71.70 | 71.16 |
| MaPLe (Khattak et al., 2023a) | | 68.73 | 93.67 | 91.53 | 68.07 | 78.80 | 84.40 | 27.97 | 68.40 | 50.97 | 41.90 | 72.60 | 67.91 |
| PromptSRC (Khattak et al., 2023b) | | 68.13 | 93.67 | 92.00 | 69.40 | 85.93 | 84.87 | 27.67 | 69.67 | 56.23 | 73.13 | 74.80 | 72.32 |
| AMU-Tuning (Tang et al., 2024) | | 67.37 | 94.51 | 91.01 | 65.65 | 80.29 | 85.15 | 27.20 | 65.11 | 51.77 | 70.76 | 71.89 | 70.07 |
| TCP (Yao et al., 2024) | 1-shot | 67.03 | 93.53 | 91.43 | 66.17 | 86.87 | 84.67 | 28.87 | 69.00 | 54.80 | 62.53 | 73.37 | 70.75 |
| DePT (Zhang et al., 2024) | | 64.23 | 93.70 | 90.43 | 67.63 | 82.77 | 84.70 | 30.03 | 68.03 | 54.07 | 52.30 | 74.03 | 69.27 |
| GalLop (Lafon et al., 2024) | | 69.79 | 94.11 | 91.59 | **71.47** | 86.12 | 84.81 | **32.52** | 68.82 | 57.15 | 63.58 | 73.11 | 72.10 |
| TextRefiner (Xie et al., 2024) | | 69.00 | 92.13 | 88.17 | 65.07 | 70.10 | 85.40 | 25.63 | 67.97 | 44.83 | 53.93 | 69.93 | 66.56 |
| MMRL (Guo & Gu, 2025) | | 69.00 | 94.17 | 90.87 | 68.70 | 85.97 | 84.87 | 28.53 | 68.90 | 56.37 | 76.00 | 75.97 | 72.67 |
| SkipT (Wu et al., 2025) | | 69.20 | 93.87 | 91.60 | 69.63 | 83.63 | 85.67 | 29.93 | 69.10 | 54.50 | 72.23 | 75.30 | 72.24 |
| LDC (Li et al., 2025) | | 69.54 | 93.79 | 91.25 | 68.24 | 83.64 | 85.88 | 27.57 | 67.99 | **58.22** | **78.49** | 73.20 | 72.53 |
| TAC (Hao et al., 2025) | | 69.80 | **94.60** | **92.10** | 71.33 | 85.27 | 85.70 | 31.17 | **71.17** | 56.87 | 70.47 | 76.47 | 73.18 |
| MPS-Tuning (Ours) | | **70.37** | 94.47 | 91.17 | 70.17 | 86.50 | 86.10 | 29.97 | 70.40 | 56.47 | 76.10 | **77.40** | **73.55** |
| CLIP (Radford et al., 2021) | | 66.73 | 93.35 | 88.25 | 65.48 | 67.44 | 83.65 | 23.67 | 62.59 | 44.27 | 42.01 | 65.13 | 63.87 |
| CoOp (Zhou et al., 2022b) | | 67.07 | 93.07 | 89.80 | 70.50 | 87.33 | 84.40 | 26.20 | 66.53 | 53.60 | 65.17 | 73.43 | 70.65 |
| Tip-adapter-F (Zhang et al., 2022) | | 70.04 | 94.20 | 91.47 | 70.91 | 89.65 | 86.39 | 33.51 | 68.64 | 55.91 | 73.17 | 76.10 | 73.64 |
| PLOT++ (Chen et al., 2023) | | 68.28 | 94.69 | 92.29 | 73.17 | 89.81 | 86.33 | 31.14 | 68.06 | 56.72 | 76.80 | 76.76 | 74.00 |
| TaskRes (Yu et al., 2023) | | 70.20 | 94.23 | 90.67 | 72.27 | 89.70 | 85.60 | 32.67 | 70.43 | 55.67 | 70.23 | 75.20 | 73.33 |
| MaPLe (Khattak et al., 2023a) | | 69.47 | 94.20 | 92.63 | 69.80 | 84.47 | 85.03 | 30.93 | 70.53 | 55.63 | 72.30 | 74.30 | 72.66 |
| PromptSRC (Khattak et al., 2023b) | | 69.77 | 94.53 | 92.50 | 73.40 | 91.17 | 85.70 | 31.70 | 71.60 | 59.97 | 79.37 | 78.50 | 75.29 |
| AMU-Tuning (Tang et al., 2024) | | 69.62 | 94.86 | 90.41 | 68.52 | 83.31 | 85.51 | 31.15 | 67.67 | 54.04 | 72.55 | 74.85 | 72.04 |
| TCP (Yao et al., 2024) | 2-shot | 68.30 | 94.13 | 90.57 | 71.00 | 90.87 | 85.17 | 32.23 | 71.03 | 58.43 | 70.63 | 77.70 | 73.64 |
| DePT (Zhang et al., 2024) | | 65.97 | 94.27 | 90.90 | 71.77 | 88.80 | 85.30 | 32.43 | 70.37 | 59.83 | 68.57 | 77.60 | 73.25 |
| GalLop (Lafon et al., 2024) | | 70.57 | 95.25 | **93.09** | 75.03 | **91.81** | 85.42 | **36.74** | 71.17 | 61.90 | 67.83 | 77.81 | 75.15 |
| TextRefiner (Xie et al., 2024) | | 70.10 | 93.37 | 90.10 | 65.17 | 73.37 | 85.70 | 26.63 | 69.07 | 48.70 | 53.40 | 69.77 | 67.76 |
| MMRL (Guo & Gu, 2025) | | 70.30 | 94.83 | 91.57 | 72.93 | 91.20 | 85.53 | 34.23 | 71.53 | 61.37 | **82.87** | 78.50 | 75.90 |
| SkipT (Wu et al., 2025) | | 70.23 | 95.10 | 92.43 | 73.17 | 90.33 | 85.90 | 33.53 | 70.60 | 58.83 | 78.50 | 78.37 | 75.18 |
| LDC (Li et al., 2025) | | 69.86 | 94.32 | 91.17 | 70.75 | 88.71 | 86.07 | 28.98 | 69.61 | 62.17 | 81.73 | 75.95 | 74.48 |
| TAC (Hao et al., 2025) | | 70.73 | 95.27 | 92.63 | 74.87 | 91.10 | 86.30 | 36.37 | **72.97** | 60.93 | 80.97 | 79.30 | 76.49 |
| MPS-Tuning (Ours) | | **71.43** | **95.50** | 92.40 | **75.37** | 91.57 | **86.70** | 33.77 | 72.37 | 61.47 | 81.63 | 81.03 | **76.66** |
| CLIP (Radford et al., 2021) | | 66.73 | 93.35 | 88.25 | 65.48 | 67.44 | 83.65 | 23.67 | 62.59 | 44.27 | 42.01 | 65.13 | 63.87 |
| CoOp (Zhou et al., 2022b) | | 68.73 | 94.40 | 92.57 | 74.47 | 92.17 | 84.47 | 30.83 | 69.97 | 58.70 | 70.80 | 77.10 | 74.02 |
| Tip-adapter-F (Zhang et al., 2022) | | 70.70 | 95.01 | 92.04 | 74.57 | 92.61 | 86.67 | 36.45 | 70.77 | 61.70 | 79.22 | 79.51 | 76.30 |
| PLOT++ (Chen et al., 2023) | | 70.40 | 95.13 | 92.55 | 76.25 | 92.93 | 86.46 | 35.29 | 71.73 | 62.43 | 83.21 | 79.76 | 76.92 |
| TaskRes (Yu et al., 2023) | | 70.93 | 95.00 | 91.93 | 75.97 | 91.73 | 86.03 | 33.40 | 72.70 | 60.17 | 74.17 | 76.20 | 75.29 |
| MaPLe (Khattak et al., 2023a) | | 70.77 | 95.30 | 93.27 | 71.97 | 90.47 | 85.67 | 32.63 | 72.73 | 61.17 | 76.57 | 77.93 | 75.32 |
| PromptSRC (Khattak et al., 2023b) | | 71.07 | 95.27 | 93.43 | 77.13 | 93.87 | 86.17 | 37.47 | 74.00 | 65.53 | 86.30 | 81.57 | 78.35 |
| AMU-Tuning (Tang et al., 2024) | | 71.69 | 96.21 | 92.31 | 71.52 | 86.28 | 85.77 | 34.48 | 70.24 | 59.32 | 78.63 | 76.67 | 74.83 |
| TCP (Yao et al., 2024) | 4-shot | 69.97 | 95.17 | 92.00 | 76.43 | 93.83 | 85.50 | 36.37 | 73.40 | 64.07 | 73.63 | 80.77 | 76.47 |
| DePT (Zhang et al., 2024) | | 68.57 | 95.10 | 92.83 | 76.13 | 93.87 | 85.70 | 34.57 | 73.07 | 64.13 | 78.87 | 81.03 | 76.72 |
| GalLop (Lafon et al., 2024) | | 71.67 | 95.60 | 93.22 | 79.09 | 95.58 | 86.08 | **42.65** | 73.42 | 66.88 | 74.05 | 81.23 | 78.19 |
| TextRefiner (Xie et al., 2024) | | 70.70 | 93.13 | 92.57 | 67.90 | 76.17 | 87.03 | 29.43 | 70.80 | 51.37 | 56.00 | 71.87 | 69.72 |
| MMRL (Guo & Gu, 2025) | | 71.40 | 96.03 | 92.57 | 78.17 | 94.60 | 85.77 | 40.47 | 73.93 | 67.87 | 87.67 | 82.67 | 79.20 |
| SkipT (Wu et al., 2025) | | 71.40 | 95.60 | 93.33 | 77.60 | 94.27 | 86.00 | 39.90 | 73.07 | 65.70 | 83.40 | 82.53 | 78.44 |
| LDC (Li et al., 2025) | | 71.04 | 95.25 | 91.80 | 75.13 | 93.95 | 86.71 | 31.92 | 72.92 | 66.43 | 86.37 | 79.75 | 77.39 |
| TAC (Hao et al., 2025) | | 71.73 | 95.60 | **93.67** | 78.63 | 94.50 | 86.80 | 41.93 | **74.70** | 66.67 | **88.70** | 82.93 | 79.62 |
| MPS-Tuning (Ours) | | **72.57** | **96.80** | **93.67** | **80.47** | **96.00** | **87.17** | 41.97 | 74.53 | **68.50** | 88.50 | 84.30 | **80.47** |
| CLIP (Radford et al., 2021) | | 66.73 | 93.35 | 88.25 | 65.48 | 67.44 | 83.65 | 23.67 | 62.59 | 44.27 | 42.01 | 65.13 | 63.87 |
| CoOp (Zhou et al., 2022b) | | 70.63 | 94.37 | 91.27 | 79.30 | 94.97 | 82.67 | 39.00 | 71.53 | 64.77 | 78.07 | 80.20 | 76.98 |
| Tip-adapter-F (Zhang et al., 2022) | | 72.01 | 95.17 | 91.88 | 77.91 | 94.88 | 86.80 | 41.94 | 73.93 | 67.91 | 84.35 | 82.37 | 79.01 |
| PLOT++ (Chen et al., 2023) | | 71.31 | 95.51 | 93.02 | 81.26 | 95.44 | 86.58 | 41.42 | 73.93 | 66.49 | 88.37 | 82.80 | 79.65 |
| TaskRes (Yu et al., 2023) | | 72.20 | 95.30 | 92.00 | 79.60 | 96.70 | 86.40 | 40.27 | 74.57 | 66.60 | 77.47 | 81.67 | 78.43 |
| MaPLe (Khattak et al., 2023a) | | 71.63 | 95.40 | 93.07 | 74.63 | 94.27 | 86.47 | 36.97 | 74.00 | 65.57 | 84.60 | 81.03 | 77.97 |
| PromptSRC (Khattak et al., 2023b) | | 72.33 | 95.67 | 93.50 | 80.97 | 96.27 | 86.90 | 43.27 | 75.73 | 69.87 | 88.80 | 84.30 | 80.69 |
| AMU-Tuning (Tang et al., 2024) | | 73.56 | 96.74 | 93.34 | 74.62 | 89.38 | 86.04 | 38.86 | 71.32 | 61.39 | 80.78 | 79.59 | 76.87 |
| TCP (Yao et al., 2024) | 8-shot | 71.60 | 95.43 | 92.30 | 80.27 | 96.20 | 86.53 | 40.93 | 75.07 | 68.80 | 77.57 | 83.23 | 78.90 |
| DePT (Zhang et al., 2024) | | 71.37 | 95.70 | 93.47 | 80.63 | 96.27 | 86.60 | 41.00 | 75.30 | 69.40 | 82.83 | 83.57 | 79.65 |
| GalLop (Lafon et al., 2024) | | 72.86 | 96.19 | 93.73 | 84.42 | 97.90 | 86.79 | 48.48 | 75.51 | 71.89 | 84.04 | 84.40 | 81.48 |
| TextRefiner (Xie et al., 2024) | | 70.77 | 95.17 | 92.70 | 69.87 | 84.07 | 86.97 | 31.00 | 72.17 | 55.23 | 59.57 | 75.37 | 72.08 |
| MMRL (Guo & Gu, 2025) | | 72.33 | 96.27 | 93.03 | 82.57 | 96.60 | 86.33 | 48.07 | 76.00 | 71.60 | 88.73 | 84.67 | 81.47 |
| SkipT (Wu et al., 2025) | | 72.40 | 95.90 | 93.40 | 82.33 | 96.60 | 86.73 | 46.50 | 74.77 | 69.77 | 86.57 | 85.60 | 80.96 |
| LDC (Li et al., 2025) | | 72.48 | 95.86 | 92.48 | 79.69 | 95.98 | 86.94 | 37.98 | 75.56 | 71.51 | 90.80 | 80.91 | 80.02 |
| TAC (Hao et al., 2025) | | 72.73 | 96.27 | **94.20** | 82.43 | 96.77 | 87.20 | 48.63 | 76.17 | 70.73 | 89.63 | 85.83 | 81.85 |
| MPS-Tuning (Ours) | | **74.07** | **96.83** | 93.77 | **87.20** | **98.03** | **87.63** | **55.60** | **76.70** | **73.30** | **91.13** | **87.03** | **83.75** |
| CLIP (Radford et al., 2021) | | 66.73 | 93.35 | 88.25 | 65.48 | 67.44 | 83.65 | 23.67 | 62.59 | 44.27 | 42.01 | 65.13 | 63.87 |
| CoOp (Zhou et al., 2022b) | | 71.87 | 95.57 | 91.87 | 83.07 | 97.07 | 84.20 | 43.40 | 74.67 | 69.87 | 84.93 | 82.23 | 79.89 |
| Tip-adapter-F (Zhang et al., 2022) | | 73.71 | 96.11 | 93.02 | 84.55 | 96.26 | 87.34 | 45.03 | 76.30 | 72.46 | 88.47 | 85.12 | 81.59 |
| PLOT++ (Chen et al., 2023) | | 72.60 | 96.04 | 93.59 | 84.55 | 97.56 | 87.11 | 46.74 | 76.03 | 71.43 | 92.00 | 85.34 | 82.09 |
| TaskRes (Yu et al., 2023) | | 73.03 | 95.80 | 92.40 | 83.47 | 97.93 | 86.90 | 44.90 | 76.07 | 71.57 | 82.70 | 83.97 | 80.79 |
| MaPLe (Khattak et al., 2023a) | | 72.57 | 95.67 | 93.30 | 78.00 | 95.80 | 87.37 | 40.63 | 75.47 | 70.50 | 89.03 | 82.70 | 80.09 |
| PromptSRC (Khattak et al., 2023b) | | 73.17 | 96.07 | 93.67 | 83.83 | 97.60 | 87.50 | 50.83 | 77.23 | 72.73 | 92.43 | 86.47 | 82.87 |
| AMU-Tuning (Tang et al., 2024) | | 74.93 | 97.04 | 93.46 | 78.10 | 90.95 | 86.40 | 43.12 | 72.78 | 64.66 | 82.14 | 80.54 | 78.56 |
| TCP (Yao et al., 2024) | 16-shot | 72.40 | 95.83 | 92.77 | 84.00 | 97.43 | 87.23 | 46.13 | 76.80 | 72.70 | 85.00 | 85.40 | 81.43 |
| DePT (Zhang et al., 2024) | | 73.35 | 96.27 | 94.13 | 85.03 | 97.83 | 87.30 | 47.00 | 77.30 | 74.03 | 91.43 | 85.57 | 82.66 |
| GalLop (Lafon et al., 2024) | | 74.11 | 96.64 | 94.30 | 88.24 | 98.67 | 87.42 | 55.16 | 77.25 | 75.00 | 89.24 | 86.64 | 83.88 |
| TextRefiner (Xie et al., 2024) | | 70.90 | 95.27 | 93.13 | 72.10 | 92.37 | 87.23 | 34.37 | 74.33 | 61.93 | 64.30 | 79.63 | 75.05 |
| MMRL (Guo & Gu, 2025) | | 73.40 | 97.13 | 93.83 | 86.43 | 98.40 | 87.03 | 57.60 | 77.70 | 75.30 | 93.37 | 87.60 | 84.34 |
| SkipT (Wu et al., 2025) | | 73.83 | 96.77 | 94.00 | 87.43 | 98.47 | 87.13 | 55.57 | 76.80 | 74.07 | 91.57 | 87.70 | 83.91 |
| LDC (Li et al., 2025) | | 73.88 | 96.63 | 93.35 | 84.11 | 97.85 | 87.31 | 47.88 | 76.91 | 77.07 | 92.16 | 84.46 | 82.87 |
| TAC (Hao et al., 2025) | | 73.67 | 96.83 | 94.57 | 85.83 | 98.20 | 87.70 | 55.87 | 77.67 | 74.80 | 93.53 | 88.20 | 84.26 |
| MPS-Tuning (Ours) | | **75.60** | **97.37** | **94.77** | **91.13** | **99.37** | **88.13** | **69.03** | **78.47** | **77.20** | **94.37** | **89.87** | **86.85** |

AMU-Tuning (Tang et al., 2024), TCP (Yao et al., 2024), DePT (Zhang et al., 2024), GalLop (Lafon et al., 2024), TextRefiner (Xie et al., 2024), MMRL (Guo & Gu, 2025), SkipT (Wu et al., 2025), LDC (Li et al., 2025), and TAC (Hao et al., 2025). Our approach achieved the highest average performance under all few-shot settings (1, 2, 4, 8, and 16 shots), with its advantage becoming increasingly pronounced as more samples were introduced, underscoring its robust learning capability.

## F.2 ABLATION STUDY

Additional ablation results for the 1-, 2-, 4-, 8-, and 16-shot settings are provided in Tab. 10, and the overall trends are consistent with the main paper. A notable exception is the 1 shot case, where HMS alone does not improve performance. This likely stems from the extremely limited positive sample pool, containing only the sample itself, its augmentations, and the associated textual feature, which encourages overfitting and weakens generalization. With more shots, the increased sample diversity allows HMS to operate effectively and enhance performance.

Table 10: More ablation results under 1, 2, 4, 8, and 16-shot settings. The best result in each setting is highlighted in bold.

| Method | Setting | ImageNet | Caltech101 | OxfordPets | StanfordCars | Flowers102 | Food101 | FGVCAircraft | SUN397 | DTD | EuroSAT | UCF101 | Average |
|---|---|---|---|---|---|---|---|---|---|---|---|---|---|
| CE | | 68.80 | 94.00 | 90.13 | 68.03 | 86.50 | 84.93 | 28.17 | 69.40 | **57.53** | 74.47 | 76.63 | 72.60 |
| CE+MAR | 1-shot | 70.30 | 94.07 | 91.10 | **70.73** | **86.93** | 85.80 | **30.60** | 70.20 | 56.40 | 75.73 | 76.93 | 73.53 |
| CE+HMS | | 69.33 | 94.07 | 90.27 | 66.93 | 85.80 | 84.97 | 27.33 | 69.63 | 56.57 | 74.23 | 76.73 | 72.35 |
| MPS-Tuning | | **70.37** | **94.47** | **91.17** | 70.17 | 86.50 | **86.10** | 29.97 | **70.40** | 56.47 | **76.10** | **77.40** | **73.55** |
| CE | | 69.63 | 95.27 | 91.30 | 72.53 | **91.67** | 85.60 | 30.90 | 71.07 | **61.97** | 81.00 | 80.20 | 75.58 |
| CE+MAR | 2-shot | 71.17 | 94.97 | 92.27 | 75.30 | 91.43 | 86.40 | **34.17** | 72.07 | 61.17 | **82.03** | 80.43 | 76.49 |
| CE+HMS | | 70.33 | 95.23 | 91.33 | 73.77 | 91.00 | 85.50 | 32.93 | 71.10 | 61.63 | 80.97 | 80.37 | 75.83 |
| MPS-Tuning | | **71.43** | **95.59** | **92.40** | **75.37** | 91.57 | **86.70** | 33.77 | **72.37** | 61.47 | 81.63 | **81.03** | **76.66** |
| CE | | 70.43 | 96.33 | 92.33 | 78.30 | 95.53 | 85.73 | 37.10 | 73.10 | 68.10 | 87.83 | 83.27 | 78.92 |
| CE+MAR | 4-shot | 72.23 | 96.13 | 93.23 | 80.70 | 95.90 | 87.00 | 41.07 | 74.23 | 68.03 | 87.90 | 83.97 | 80.04 |
| CE+HMS | | 71.57 | 96.53 | 92.70 | 79.87 | 95.13 | 85.47 | 40.80 | 73.33 | 68.00 | 88.53 | 83.63 | 79.60 |
| MPS-Tuning | | **72.57** | **96.80** | **93.67** | **81.17** | **96.00** | **87.17** | **41.97** | **74.53** | **68.50** | **88.50** | **84.30** | **80.47** |
| CE | | 71.90 | 96.73 | 92.50 | 85.17 | 97.75 | 86.20 | 51.63 | 75.17 | 72.30 | 90.13 | 86.10 | 82.32 |
| CE+MAR | 8-shot | 73.67 | 96.30 | 93.47 | 86.70 | 97.70 | 87.43 | 54.80 | 76.20 | 72.33 | 90.03 | 86.40 | 83.18 |
| CE+HMS | | 73.17 | 96.50 | 93.07 | 86.60 | 97.47 | 85.87 | 55.20 | 75.93 | 72.67 | 90.83 | 86.43 | 83.07 |
| MPS-Tuning | | **74.07** | **96.83** | **93.77** | **87.20** | **98.03** | **87.63** | **55.60** | **76.70** | **73.30** | **91.13** | **87.03** | **83.75** |
| CE | | 72.93 | 97.07 | 93.53 | 90.00 | 99.02 | 86.20 | 66.33 | 76.30 | 75.87 | 93.77 | 88.43 | 85.41 |
| CE+MAR | 16-shot | 75.30 | 97.00 | 94.27 | 90.80 | 99.23 | 88.03 | 68.47 | 78.07 | 77.27 | 93.47 | 88.90 | 86.44 |
| CE+HMS | | 74.77 | **97.33** | 93.73 | 90.77 | 99.23 | 86.20 | 68.87 | 77.80 | 76.40 | 93.90 | 89.23 | 86.20 |
| MPS-Tuning | | **75.60** | 97.37 | **94.77** | **91.13** | **99.37** | **88.13** | **69.03** | **78.47** | **77.20** | **94.37** | **89.87** | **86.85** |

Table 11: Sensitivity Study on MAR weight $\lambda_1$

| $\lambda_1$ | ImageNet | Caltech101 | OxfordPets | StanfordCars | Flowers102 | Food101 | FGVCAircraft | SUN397 | DTD | EuroSAT | UCF101 | Average |
|---|---|---|---|---|---|---|---|---|---|---|---|---|
| 0.01 | 74.80 | 97.33 | 93.67 | 90.70 | 99.23 | 86.27 | 68.43 | 77.87 | 76.53 | 94.20 | 89.17 | 86.20 |
| 0.1 | 74.97 | **97.43** | 94.33 | 90.97 | 99.33 | 87.13 | 68.60 | 78.07 | 76.97 | 94.40 | 89.67 | 86.53 |
| 0.2 | 74.97 | **97.43** | 94.37 | 90.90 | 99.33 | 87.20 | 68.83 | 78.07 | 77.10 | **94.43** | 89.80 | 86.58 |
| 0.5 | 75.23 | **97.43** | 94.53 | **91.13** | 99.37 | 87.70 | **69.03** | 78.37 | 77.20 | 94.37 | **89.87** | 86.74 |
| 1 | 75.43 | **97.43** | 94.70 | 90.97 | **99.40** | 88.00 | 68.83 | 78.43 | **77.30** | 93.37 | 89.63 | **86.76** |
| 2 | **75.60** | 97.23 | **94.77** | 90.77 | 99.20 | **88.13** | 68.23 | **78.47** | 76.93 | 92.10 | 88.90 | 86.39 |
| 5 | 75.47 | 96.93 | 94.37 | 90.03 | 98.83 | 88.03 | 66.50 | 78.13 | 76.33 | 91.30 | 88.20 | 85.83 |
| 10 | 75.13 | 96.73 | 94.13 | 88.30 | 98.53 | 88.00 | 64.43 | 77.90 | 76.07 | 91.27 | 87.47 | 85.27 |

Table 12: Sensitivity Study on HMS weight $\lambda_2$

| | ImageNet | Caltech101 | OxfordPets | StanfordCars | Flowers102 | Food101 | FGVCAircraft | SUN397 | DTD | EuroSAT | UCF101 | Average |
|---|---|---|---|---|---|---|---|---|---|---|---|---|
| 0.01 | 75.37 | 97.13 | 94.37 | 91.03 | 99.27 | 88.03 | 68.93 | 78.10 | **77.20** | 93.73 | 89.17 | 86.58 |
| 0.1 | 75.60 | 97.37 | 94.77 | **91.13** | **99.37** | **88.13** | **69.03** | 78.47 | **77.20** | 94.37 | **89.87** | **86.85** |
| 0.2 | **75.63** | **97.47** | **94.87** | 91.07 | 99.20 | 88.07 | 66.73 | **78.50** | 76.93 | 94.33 | 89.77 | 86.60 |
| 0.3 | 75.60 | 97.27 | 94.73 | 91.00 | 99.13 | 87.87 | 66.93 | 78.30 | 76.97 | **94.40** | 89.63 | 86.53 |
| 0.5 | 75.50 | 95.27 | 94.57 | 90.67 | 99.10 | 87.07 | 67.63 | 77.77 | 76.80 | 94.27 | 89.33 | 86.18 |
| 0.8 | 74.90 | 93.70 | 94.13 | 88.73 | 98.80 | 86.57 | 63.37 | 66.33 | 75.77 | 93.63 | 88.23 | 84.02 |
| 1 | 74.90 | 96.37 | 93.93 | 88.17 | 98.27 | 86.33 | 50.83 | 67.87 | 75.83 | 94.17 | 88.07 | 83.16 |
| 2 | 69.07 | 91.93 | 93.27 | 80.03 | 86.60 | 84.77 | 24.43 | 64.53 | 43.20 | 87.30 | 83.83 | 73.82 |

Table 13: Sensitivity Study on logits weight $\alpha$

| | ImageNet | Caltech101 | OxfordPets | StanfordCars | Flowers102 | Food101 | FGVCAircraft | SUN397 | DTD | EuroSAT | UCF101 | Average |
|---|---|---|---|---|---|---|---|---|---|---|---|---|
| 0.1 | 74.07 | 96.97 | 94.43 | 88.30 | 95.50 | 87.90 | 58.33 | 75.60 | 71.47 | 91.70 | 87.60 | 83.81 |
| 0.2 | 75.43 | 97.27 | **94.77** | 90.83 | 99.23 | 88.10 | 68.00 | 78.07 | 76.83 | 93.87 | 89.40 | 86.53 |
| 0.3 | 75.60 | 97.37 | **94.77** | **91.13** | **99.37** | **88.13** | 69.03 | 78.47 | 77.20 | **94.37** | 89.87 | **86.85** |
| 0.4 | **75.67** | 97.40 | 94.63 | **91.13** | **99.37** | 88.07 | 68.87 | 78.57 | **77.30** | 94.20 | **89.90** | 86.83 |
| 0.5 | 75.57 | 97.37 | 94.57 | **91.13** | 99.33 | 87.87 | 68.97 | **78.67** | 77.17 | 94.17 | 89.80 | 86.78 |
| 0.6 | 75.50 | 97.33 | 94.30 | 90.97 | 99.27 | 87.63 | 68.87 | 78.60 | 77.17 | 94.00 | 89.57 | 86.65 |
| 0.7 | 75.33 | 97.40 | 93.63 | 90.70 | 99.13 | 87.07 | **69.17** | 78.50 | 76.60 | 94.13 | 89.27 | 86.45 |
| 0.8 | 75.00 | **97.47** | 92.97 | 89.80 | 98.97 | 86.33 | 67.90 | 78.27 | 76.40 | 93.20 | 88.93 | 85.93 |
| 0.9 | 74.57 | 97.20 | 91.37 | 89.33 | 99.00 | 84.93 | 67.73 | 77.83 | 76.63 | 93.00 | 88.57 | 85.47 |
| 1.0 | 74.00 | 97.07 | 89.00 | 88.33 | 98.80 | 83.03 | 67.10 | 77.20 | 75.87 | 93.37 | 87.87 | 84.69 |

## G  SENSITIVITY STUDY

We conducted sensitivity analyses for the weights of MAR ($\lambda_1$), HMS ($\lambda_2$), and the logits from the fine-tuning branch ($\alpha$), with the corresponding results shown in Tables Tab. 11, Tab. 12, and Tab. 13, respectively. Regarding $\lambda_1$, we found that setting it to 0.5 or 1 yields the best performance when a fixed value is applied across all datasets. To further improve performance, we divided the datasets into two groups and set $\lambda_1$ to 0.5 and 2 for each group (see Hyperparameter Settings), respectively.

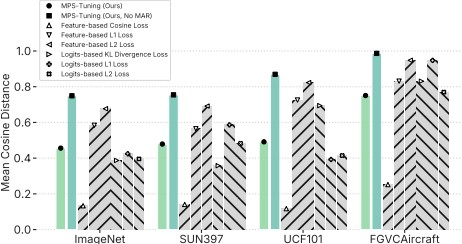

(a) Cosine-Based quantification of feature shift. A lower score indicates a stricter constraint on the output features.

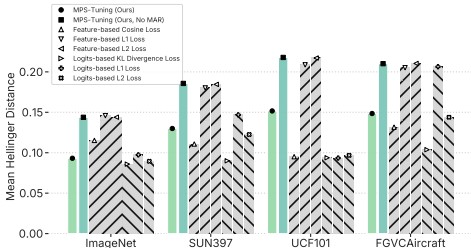

(b) Hellinger-based metric for quantifying logits shift. A lower score indicates a stricter constraint on the predicted outcomes.

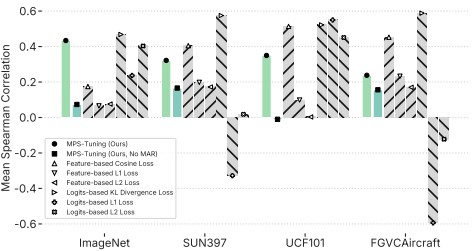

(c) Spearman-based metric for quantifying logits shift. A higher score indicates a stricter constraint on the predicted outcomes.

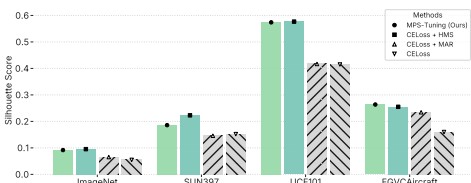

(d) Impact of MAR and HMS on feature discrimination using Silhouette coefficient. A higher score indicates better clustering performance.

Figure 10: Comparison of different metrics and constraints: (a) Cosine, (b) Hellinger, (c) Spearman, and (d) Silhouette.

As for $\lambda_2$, we fixed it at 0.1 across all datasets, and similarly, $\alpha$ was set to 0.3 for all datasets. Notably, the model's performance exhibited minimal variation when these hyperparameters were adjusted around their default values, demonstrating the robustness of our method to hyperparameter choices.

Table 14: Training cost under each method's default settings on SUN397.

|  | Time(min) | Params(M) | Mem(GB) |
|---|---|---|---|
| CoOp | 170.4 | 0.008 | 7.7 |
| Tip-Adapter | 9.4 | 3.3 | 2.9 |
| MaPLe | 13.6 | 1.2 | 7.2 |
| TCP | 21.8 | 0.3 | 9.0 |
| CoCoOp | 252.4 | 0.04 | 7.1 |
| MPS-Tuning (Ours) | 54.23 | 31 | 9.3 |

## H  TRAINING COST

Due to the fixed text encoder design, our method achieves high training efficiency with GPU memory usage and time costs comparable to previous methods (Tab. 14).

## I   MORE INTERPRETATION RESULTS

### I.1   IMPACT OF CONSISTENCY CONSTRAINTS ON MODEL ADAPTATION

To substantiate the potential limitations of different consistency constraints discussed in the interpretability section of the main text, we conducted quantitative experiments analyzing changes in feature representations and logits distributions before and after fine-tuning.

**1. Feature-Level Analysis**   We evaluated the cosine distance between model output features before and after fine-tuning across multiple datasets. In CLIP's normalized feature space used for similarity computation, the difference between two vectors is measured by their angular separation. Since angular alignment is equivalent to vector alignment in this context, a larger cosine distance between pre- and post-fine-tuning vectors indicates greater overall divergence. As shown in Fig. 10a, cosine similarity constraints result in minimal angular changes between pre- and post-fine-tuning models, even in cross-domain scenarios (e.g., FGVCAircraft, where models typically require substantial adjustments for effective downstream task adaptation). This demonstrates that cosine similarity imposes more stringent restrictions on feature variations compared to other feature-based constraints.

**2. Logits-Level Analysis**   We assessed the variation in prediction probabilities using Hellinger Distance and the consistency of prediction rankings via Spearman's Rank Correlation, comparing models before and after fine-tuning. The former quantifies differences between two probability distributions, while the latter assesses the extent to which fine-tuned models maintain prediction ranking consistency with original models. The experimental results documented in Fig. 10b and Fig. 10c show that models constrained by feature-level cosine similarity and logits-level KL divergence demonstrate markedly superior alignment with original model predictions compared to their L1 and L2 constrained counterparts. This superiority manifests through consistently reduced Hellinger Distance values and elevated Spearman's Rank Correlation coefficients, with the most pronounced differentiation occurring on the cross-domain FGVCAircraft dataset. These results confirm that cosine similarity and KL divergence heavily constrain model predictions.

These findings indicate that cosine similarity and KL divergence may impose overly rigid constraints, potentially hampering the model's learning capacity. In contrast, our proposed Manifold Alignment Regularization (MAR) offers a dynamic constraint mechanism that adapts across datasets, enabling models to engage in further learning when knowledge acquisition is necessary while effectively preserving pre-trained knowledge when high consistency exists between pre-trained and downstream task-required knowledge, thereby significantly enhancing model learning capability.

### I.2   IMPACT OF MAR AND HMS ON FEATURE DISCRIMINATION

We further employed a common clustering metric, the Silhouette Coefficient, to evaluate the class separability of features in the representation space across different model components. As shown in Fig. 10d, using MAR alone improves class separability in certain scenarios, while HMS alone significantly enhances the distinction between categories. The combined use of MAR and HMS yields results comparable to using HMS alone, indicating the effectiveness of our approach in facilitating discriminative feature learning.

### I.3   COMPARISON OF MANIFOLD ALIGNMENT PERFORMANCE

To rigorously verify manifold-level alignment, we employed Topological Data Analysis (TDA) via Persistent Homology to quantify the topological consistency between the original CLIP manifold and those generated by different fine-tuning methods. Specifically, we calculated the Wasserstein distance on persistence diagrams for $H_0$ (connected components, reflecting macro-separability) and $H_1$ (loops, reflecting fine-grained geometry) to measure structural deviation. Empirical results (Fig. 11) across multiple datasets demonstrate that MPS-Tuning achieves consistently lower Wasserstein distances compared to baselines, indicating superior capability in preserving topological structures. It is important to note that while we do not claim strict homeomorphism or homology guarantees, which are intractable in deep representation learning, the TDA evidence confirms that

our regularization effectively preserves key homological features of the original manifold and prevents structural distortion beyond simple semantic approximation.

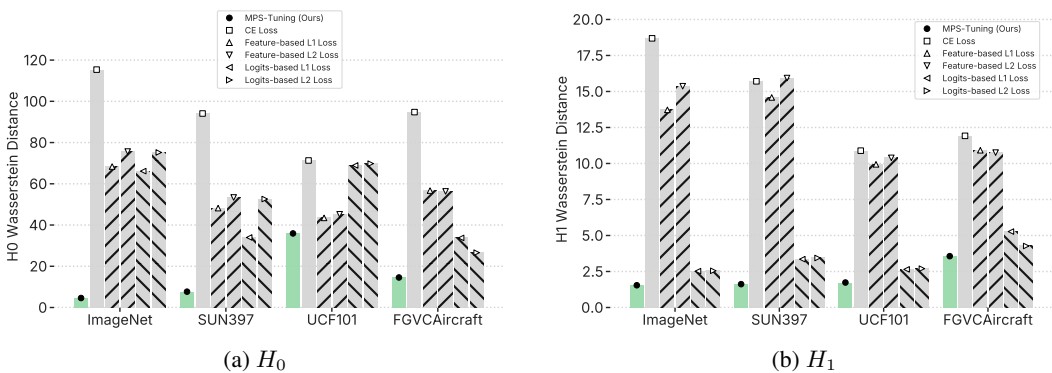

(a) $H_0$                     (b) $H_1$

Figure 11: A comparison of manifold alignment performance is conducted using TDA analysis, where lower numerical values indicate better manifold alignment.

## J  VISUALIZATION

We visualize the features using t-SNE on the ImageNet, StanfordCars, and FGVCAircraft datasets under the 16-shot setting. As illustrated in Fig. 12, MPS-Tuning yields superior intra-class compactness and inter-class separability compared to both the non-fine-tuned model and Cross-Entropy loss (CEloss) fine-tuning. Notably, in scenarios where the original CLIP model performs well (e.g., ImageNet and StanfordCars), MPS-Tuning more effectively preserves the original semantic structure and inter-class relationships than CE loss tuning. In contrast, for datasets where CLIP underperforms (e.g., FGVCAircraft), the global semantic structure is largely retained, while local adjustments facilitate improved classification.

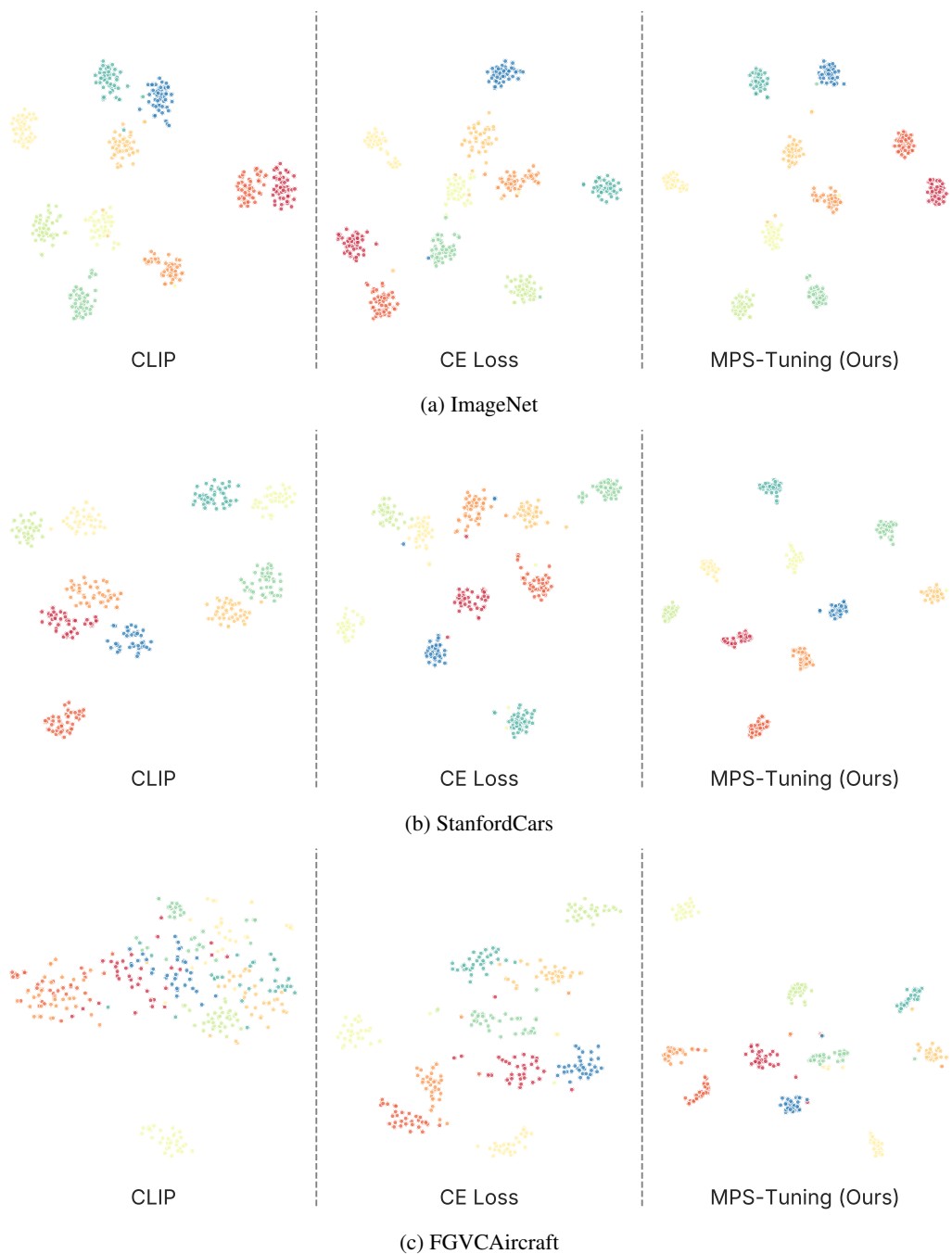

Figure 12: The t-SNE visualization, with each color denoting a distinct class.

