# OpenReview forum: "Preserve and Sculpt: Manifold-Aligned Fine-tuning of Vision-Language Models for Few-Shot Learning"
_ICLR.cc/2026/Conference — ICLR 2026 Poster_

### Official Review · Reviewer_uDYd · 2025-10-25

**Soundness:** 2
**Presentation:** 3
**Contribution:** 3
**Rating:** 4
**Confidence:** 5

**Summary:**

The paper introduces MPS-Tuning, a novel fine-tuning strategy for vision-language models (VLMs) like CLIP in few-shot settings. It consists of two key components:
1. Manifold Alignment Regularization (MAR): aligns Gram matrices between pretrained and fine-tuned representations to preserve both global and local manifold geometry, providing a tractable upper bound of the Gromov–Wasserstein distance.
2. Hierarchical Manifold Sculpting (HMS): enhances intra-class compactness and inter-class separation via multimodal query-support matching, extended to intermediate layers using a “pseudo-forward” projection.
Extensive experiments on 11 datasets show consistent improvements over state-of-the-art baselines, demonstrating better structure preservation and generalization.

**Strengths:**

1. Clear motivation and theoretical foundation: connects manifold preservation with Gromov–Wasserstein distance, offering a principled justification.
2. Gram-based alignment is simple yet theoretically grounded; HMS integrates seamlessly into VLM pipelines.
3. Strong and consistent gains on 11 benchmarks and two domain generalization datasets.

**Weaknesses:**

1. The paper positions itself as introducing a “manifold-preserving” paradigm, but the actual mechanism Gram matrix regularization is already well explored in representation learning and knowledge distillation. The conceptual framing feels more like re-packaging than a genuine shift in understanding.
1. The GW upper-bound proof relies on a fixed one-to-one coupling; lacks tightness or empirical verification against true GW distance.
2. Computational cost not deeply analyzed: local Gram alignment (O(N·M²)) could be expensive; missing runtime/memory breakdown.

**Questions:**

1. The ablation studies isolate MAR and HMS but do not show results for MPS-Tuning without α-fusion or with only global alignment. Would these results change the conclusion about the necessity of each component?
2. How is the proposed “manifold preservation” paradigm fundamentally different from existing Gram-based alignment regularizations?

---

> ### Author Response · Authors · 2025-11-20
> **Rebuttal by Authors**
>
> Thank you for your detailed review and insightful comments! Please allow us to address your concerns below.
>
> **W1: About the concept positioning**
>
> Although MAR shares a superficial resemblance to traditional Gram-based regularization, its theoretical motivation, application context, and implementation differ substantially. MAR marks the first introduction of a manifold alignment strategy based on Gromov--Wasserstein distance matrices within the field of VLM adaptation. Our objective is to preserve the semantic manifold of the pretrained model by minimizing the GW distance, and to ensure computational feasibility, we demonstrate that aligning the p-norm difference of Gram matrices serves as an approximate upper bound to the p-order GW distance. This derivation offers a novel theoretical perspective that connects Gram-based regularization with GW distance, providing a rigorous foundation for our method while simultaneously offering a theoretical explanation for prior heuristic Gram-based approaches. Furthermore, unlike existing regularizers used for knowledge transfer between heterogeneous models of different architectures or scales, MAR is designed to retain pretrained knowledge within the same model architecture during fine-tuning. Finally, MAR performs manifold alignment across multiple representational scales instead of only enforcing global sample level alignment as in conventional Gram-based approaches [1,2], which enables effectiveness across diverse scenarios as demonstrated in our ablation studies.
>
> *[1] Park, Wonpyo, et al. "Relational knowledge distillation." CVPR. 2019.*
>
> *[2]Peng, Baoyun, et al. "Correlation congruence for knowledge distillation." ICCV. 2019.*
>
> **W2: About tightness analysis**
>
> To validate the tightness of our derived upper bound for the Gromov--Wasserstein distance, we compared our bound against the numerical GW distance calculated via the Python Optimal Transport (POT) library during the training phase. This comparison was conducted on the test sets of both natural image datasets (ImageNet, SUN397) and cross-domain datasets (FGVCAircraft, UCF101), measuring the distance between fine-tuned and original CLIP features. As detailed in the Section 4.5 of the revised manuscript , our upper bound closely tracks the POT-calculated values across all benchmarks. These empirical results confirm that our method provides a theoretically tight approximation effectively serving the VLM adaptation task.
>
> **W3: About computational cost**
>
> As shown in the table below, we provide a statistical analysis of the computational overhead. By freezing the text encoder, our method achieves superior performance while maintaining GPU memory usage and training duration comparable to baseline methods, thereby demonstrating high training efficiency.
>
> **Training cost under each method's default settings on SUN397**
> ||**Time(min)**|**Params(M)**|**Mem(GB)**|
> |:---|:---:|:---:|:---:|
> |CoOp|170.4|0.008|7.7|
> |Tip-Adapter|9.4|3.3|2.9|
> |MaPLe|13.6|1.2|7.2|
> |TCP|21.8|0.3|9.0|
> |CoCoOp|252.4|0.04|7.1|
> |MPS-Tuning (Ours)|54.23|31|9.3|
>
> **Q1: Concern on sufficiency of ablation results**
>
> Our Table 4 in the main text reports ablation results on MAR, isolating the effects of global and local alignment. The results show that each alignment mechanism excels under different scenarios, while their combination yields the best overall performance. In the supplementary material, Table 13 in Appendix presents the sensitivity analysis of alpha. When alpha is set to 1, meaning that no logits reweighting is applied based on CLIP predictions, the model still achieves an SOTA accuracy of 84.69.
>
> **Q2: Difference from existing Gram-based alignment regularizations**
>
> Our implementation generalizes Gram-matrix alignment across multiple layers and demonstrates consistent gains, marking a clear departure from standard Gram-based regularizers. On the theoretical side, we are the first to show its explicit relation to the Gromov–Wasserstein distance, which rigorously supports the proposed manifold preservation mechanism. Practically, the method avoids any large-to-small knowledge distillation pipeline and instead preserves knowledge strictly within a single pretrained model during fine-tuning.

---

> > ### Comment · Reviewer_uDYd · 2025-11-27
> >
> > Thank you for the detailed rebuttal. I appreciate the clarifications, but after reviewing the new explanations, my overall assessment remains unchanged.
> >
> > While the paper provides an interesting reinterpretation of Gram-based regularization through the lens of Gromov–Wasserstein distance, the core optimization objective remains numerically very similar to existing Gram alignment methods. As such, much of the novelty appears to lie in theoretical framing.
> >
> > The rebuttal strengthens the motivation but does not fully address this conceptual concern. Therefore, I am keeping my original score.

---

> > > ### Author Response · Authors · 2025-12-01
> > >
> > > We appreciate your comment. In response to the concern about the perceived similarity between our optimization objective and existing methods, we wish to clarify that assessing novelty solely based on the underlying mathematical operator may overlook the methodological contribution. Much like how contrastive learning methods (e.g., SimCLR, CLIP) are not a repackaging of dot-product similarity but a paradigm shift in how representation structures are optimized, our work’s novelty lies not in the Gram operator itself, but in its novel context and formulation: (1) Theoretical Derivation: We provide the first rigorous proof linking Gram alignment to the Gromov-Wasserstein distance in VLM fine-tuning; (2) Hierarchical Design: We extend this to token-level constraints specifically for ViTs, unlike generic batch-level distillation; and (3) Synergy: We integrate this as a geometric anchor to enable our manifold sculpting, yielding SOTA results that standard Gram-based methods cannot achieve.

---

### Official Review · Reviewer_TNxY · 2025-10-31

**Soundness:** 2
**Presentation:** 3
**Contribution:** 2
**Rating:** 6
**Confidence:** 5

**Summary:**

This paper introduces MPS-Tuning (Manifold-Preserving and Sculpting Tuning), a fine-tuning framework for few-shot adaptation of pretrained vision-language models such as CLIP. The method treats the feature distribution as a semantic manifold and employs Manifold Alignment Regularization (MAR) to preserve its intrinsic geometry via Gram matrix alignment. Hierarchical Manifold Sculpting (HMS) refines local structures by enhancing intra-class compactness and inter-class separability across multiple layers.
Extensive experiments demonstrate that MPS-Tuning outperforms state-of-the-art methods, with its effectiveness further validated by generalization studies.

**Strengths:**

1. The paper attempts to align the semantic spaces of images and texts from a manifold perspective, and designs two complementary modules — Manifold Alignment Regularization (MAR) and Hierarchical Manifold Sculpting (HMS) — to jointly balance knowledge preservation and adaptation to new domains. It also provides a theoretical foundation for semantic alignment under the Gromov–Wasserstein (GW) constraint.

2. The paper conducts extensive experiments across multiple datasets, along with comprehensive ablation studies, to thoroughly validate the effectiveness of the proposed method.

**Weaknesses:**

1. Although the paper claims to constrain manifold alignment through the Gromov–Wasserstein (GW) distance, the metrics and formulations it employs do not appear to have an explicit mathematical correspondence to manifold geometry. Instead, the approach seems more accurately described as aligning semantic feature distributions rather than true manifolds. The use of the manifold concept thus appears somewhat overstretched.

2. Since the proposed GM alignment serves as an upper-bound approximation to the GW distance, it raises questions about potential loss of alignment fidelity—specifically, whether this surrogate can genuinely achieve manifold-level correspondence, such as homeomorphism or homology preservation, or if it merely approximates semantic similarity in feature space.

3. The proposed method is relatively complex, consisting of several loss and sub-loss functions, which leads to considerable computational overhead. Moreover, the paper does not provide strong empirical or theoretical evidence demonstrating the actual effectiveness of the Gromov–Wasserstein (GW) formulation in achieving meaningful manifold alignment.

**Questions:**

See weakness.

---

> ### Author Response · Authors · 2025-11-20
> **Rebuttal by Authors**
>
> Thank you for your detailed review and insightful comments! Please allow us to address your concerns below.
>
> **W1: About the role of manifold concept in MPS-Tuning**
>
> We would like to clarify that our use of ``Manifold Alignment" is grounded in the strict definition of discrete metric geometry. Specifically, since CLIP representations are $L_2$-normalized, they naturally reside on a unit hypersphere $\mathbb{S}^{d-1}$, which is a classical Riemannian manifold. On this manifold, the Gram matrix (composed of cosine similarities) serves as a monotonic surrogate representation of the pairwise geodesic distances (where $d_{geo} = \arccos(\langle x, y \rangle)$), thereby encoding the intrinsic geometry and isometric invariants of the data structure rather than merely statistical moments. Furthermore, as proven in Theorem 1, our optimization of the Gram matrix difference acts as a tractable upper bound for the Gromov--Wasserstein (GW) distance. In optimal transport theory, the GW distance is explicitly defined to measure the structural discrepancy between two metric spaces (manifolds) based on their intra-relational topology. Therefore, our approach mathematically minimizes the geometric distortion of the spherical manifold, distinguishing it fundamentally from standard feature distribution alignment methods that ignore these topological relationships.
>
> **W2: About manifold-level correspondence**
>
> To rigorously verify manifold-level alignment, we employed Topological Data Analysis (TDA) via Persistent Homology to quantify the topological consistency between the original CLIP manifold and those generated by different regularization methods. Specifically, we calculated the Wasserstein distance on persistence diagrams for $H_0$ (connected components, reflecting macro-separability) and $H_1$ (loops, reflecting fine-grained geometry) to measure structural deviation. Empirical results (Appendix I.3 of the revised manuscript) across multiple datasets demonstrate that MPS-Tuning achieves consistently lower Wasserstein distances compared to baselines, indicating superior capability in preserving topological structures. Notably, while we do not claim strict homeomorphism or homology guarantees, which are intractable in deep representation learning, the TDA evidence confirms that our regularization effectively preserves key homological features of the original manifold and prevents structural distortion beyond simple semantic approximation.
>
> **W3: About computational cost and manifold alignment**
>
> We have provided a detailed breakdown of computational costs in the following Table 1. Our method achieves high training efficiency with GPU memory usage and time costs comparable to previous methods, primarily due to the fixed text encoder design. On the theoretical suitability of GW distance for manifold alignment, the GW formulation explicitly targets the alignment of metric measure spaces by minimizing the difference between their intra-space distance matrices. This mechanism ensures the preservation of intrinsic geometric structures rather than relying on coordinate correspondence, thereby offering invariance to rigid transformations and sampling shifts. We validated this property through synthetic experiments comparing an S-shaped manifold $\mathcal{A}$ with its isometric copies, noisy counterparts $\mathcal{A}_{pert}$, intrinsically scaled versions $\mathcal{A'}$, and topologically different ring manifolds $\mathcal{B}$. The quantitative results in the following Table 2 demonstrate the invariance of GW metrics to isometries, their robustness to noise, and their discriminative power regarding intrinsic topological differences.
>
> **Table 1: Training cost under each method's default settings on SUN397.**
>
> ||**Time(min)**|**Params(M)**|**Mem(GB)**|
> |:---|:---:|:---:|:---:|
> |CoOp|170.4|0.008|7.7|
> |Tip-Adapter|9.4|3.3|2.9|
> |MaPLe|13.6|1.2|7.2|
> |TCP|21.8|0.3|9.0|
> |CoCoOp|252.4|0.04|7.1|
> |MPS-Tuning (Ours)|54.23|31|9.3|
>
> **Table 2: Quantitative analysis of the GW distance on synthetic manifold experiments**
> |Comparison Group|Description & Property Tested|GW Distance|
> |:---|:---|:---:|
> |$\mathcal{A}$ vs. $\mathcal{A}$|Isometric copy (Rigid Transformation Invariance)|0.0000|
> |$\mathcal{A}$ vs. $\mathcal{A}_{pert}$|Noisy counterpart (Robustness to Noise)|0.0025|
> |$\mathcal{A}$ vs. $\mathcal{A}'$|Intrinsically scaled version (Scale Sensitivity)|0.0069|
> |$\mathcal{A}$ vs. $\mathcal{B}$|Topologically different Ring (Structure Discrimination)|0.0317|

---

### Official Review · Reviewer_xocW · 2025-11-01

**Soundness:** 3
**Presentation:** 4
**Contribution:** 3
**Rating:** 8
**Confidence:** 4

**Summary:**

This paper proposes Manifold-Preserving and Sculpting Tuning (MPS-Tuning), a novel fine-tuning framework for adapting VLMs like CLIP to few-shot classification tasks. The authors' core argument is that existing methods, which rely on parameter-efficient tuning (PEFT) or point-wise consistency constraints (e.g., $L_2$ or KL loss between features), neglect the geometric structure of the pretrained feature manifold, leading to semantic distortion and overfitting. The combination of the proposed MAR (regularizer) and HMS (objective) allows for fine-tuning a substantial portion of the model while avoiding overfitting. The authors demonstrate state-of-the-art results on 11 few-shot benchmarks, with particularly strong gains in 8-shot and 16-shot settings.

**Strengths:**

Novel and Well-Motivated Conceptual Framework: The paper's primary strength is its conceptual shift from point-wise consistency to manifold-level consistency. The idea that preserving relationships between samples (in the Gram matrix) is more important than preserving the exact feature vector of each sample is intuitive and powerful.

Strong Theoretical Grounding: The connection of the MAR loss to the Gromov-Wasserstein (GW) distance (Theorem 1, Appendix B) provides a solid theoretical foundation for the method. While it relies on a simplified upper bound (using a fixed coupling), this is a standard and necessary step to make the concept computationally tractable and provides a principled justification for using Gram matrix alignment.

Excellent Empirical Results: The method achieves new state-of-the-art performance across a comprehensive suite of 11 few-shot datasets (Table 8). It outperforms a very strong and recent set of baselines (e.g., GalLop, TAC, MMRL). The fact that the performance gap widens as the shot count increases (e.g., the 16-shot average is 86.85%, a significant +2.5-3% jump over top competitors) suggests this is a very robust learning framework, not just a 1-shot trick. The SOTA domain generalization results (Table 1) further strengthen this.

Thorough and Convincing Ablation Studies: The authors have done an excellent job of validating their design choices.

**Weaknesses:**

The paper is very strong, and my points are primarily requests for clarification rather than major criticisms.

1. Justification of "Pseudo Forward" Mechanism: A key component of the "Hierarchical" sculpting is the pseudo-forward projection (Fig. 3, Eq. 10), which projects intermediate features to the output space by skipping the Attention modules but keeping the $V_{Proj}$ and $FFN$ layers. This is a very specific and unusual design.

Q1: Could the authors provide more intuition or justification for this? Why is this the correct projection? For instance, why skip Attention but keep $V_{Proj}$? Is this based on an analysis of information flow, or was it an empirical design choice that worked well?

2. Clarity on Fine-Tuning Strategy (Appendix D): The paper's method is not a standard PEFT (like LoRA or Adapter) but rather a regularizer for partial fine-tuning. Appendix D details a "hierarchical fine-tuning strategy" where the first 4 layers are frozen, the next 4 have a parallel adapter-like module, and the last 4 are fully fine-tuned. This is a non-trivial number of trainable parameters.

Q2: How critical is this specific tuning strategy to the method's success? The ablations in Table 2 all seem to use this strategy as a base. How would MPS-Tuning perform if applied to a standard full fine-tuning (FFT) baseline, or a standard PEFT method like LoRA? It is difficult to disentangle the gains from the novel regularizer (MAR+HMS) versus the gains from this specific partial-tuning strategy.

3. Complexity of HMS Loss: The Hierarchical Manifold Sculpting (HMS) loss uses a combined support set $\mathcal{S} = \mathcal{Q} \cup \mathcal{T}$, meaning it performs image-image and image-text contrastive learning simultaneously.

Q3: Was this combined support set found to be superior to a simpler approach, such as two separate losses (one for I-I alignment and one for I-T alignment)?

**Questions:**

None

---

> ### Author Response · Authors · 2025-11-20
> **Rebuttal by Authors**
>
> Thank you for your detailed review and insightful comments! Please allow us to address your concerns below.
>
> **W/Q1:Justification of "Pseudo Forward" Mechanism**
>
> In HMS, our goal is to extend manifold sculpting to intermediate layers. However, intermediate-layer features are not inherently compatible with those in the joint vision–language space. By analyzing the forward information flow, we identify that two components within the transformer block determine the feature space of input tokens: the Value projection in the attention module and the nonlinear FFN.
> Specifically, in attention mechanism, Value tokens are linearly projected by the V map and then aggregated using the QK-derived attention weights. This aggregation enables cross-token information exchange, meaning only the V map affects the feature space that the CLS token inhabits. The Q and K projections primarily shape attention weights rather than the representation space itself. The FFN further applies a nonlinear transformation that changes the token feature space. Therefore, passing intermediate-layer features through the V map followed by the FFN is sufficient to align them with the output feature space, enabling their use in HMS.
>
> **W/Q2:Clarity on Fine-Tuning Strategy**
>
> To further assess the applicability of MPS-Tuning across broader fine-tuning settings, experiments under full fine-tuning (FFT) and LoRA were conducted. The results(listed below) show that our method consistently delivers substantial performance gains across these alternative tuning paradigms(with LoRA, SOTA-comparable performance was achieved), which manifests that MPS-Tuning is broadly applicable and robust to different fine-tuning strategies.
>
> **Comparison of different tuning methods plugged w/ & w/o MPS-Tuning, *PFT* denotes our default partial fine-tuning strategy**
> |Method|Setting|ImageNet|Caltech101|OxfordPets|StanfordCars|OxfordFlowers|Food101|FGVCAircraft|SUN397|DTD|EuroSAT|UCF101|Average|
> |:-|:-:|:-:|:-:|:-:|:-:|:-:|:-:|:-:|:-:|:-:|:-:|:-:|:-:|
> |FFT|1-shot|61.27|87.80|75.07|47.33|55.07|74.20|13.60|56.93|33.30|41.40|56.03|54.73|
> |FFT+MPS-Tuning||66.67|90.97|84.23|59.17|71.27|82.87|19.07|64.13|36.50|43.47|61.77|61.83|
> |LoRA||68.37|94.33|91.60|66.27|81.97|85.83|28.97|68.00|50.90|**81.37**|74.00|71.96|
> |LoRA+MPS-Tuning||70.10|94.30|**91.83**|69.20|81.23|**86.27**|29.73|69.03|50.07|78.00|74.47|72.20|
> |PFT||68.80|94.00|90.13|68.03|**86.50**|84.93|28.17|69.40|**57.53**|74.47|76.63|72.60|
> |PFT+MPS-Tuning||**70.37**|**94.47**|91.17|**70.17**|**86.50**|86.10|**29.97**|**70.40**|56.47|76.10|**77.40**|**73.55**|
> |FFT|4-shot|61.70|88.20|72.80|49.40|66.20|76.30|15.77|59.00|35.97|51.37|56.13|57.53|
> |FFT+MPS-Tuning||66.47|92.83|84.93|60.87|80.30|83.83|20.77|64.13|43.87|54.30|64.93|65.20|
> |LoRA||69.27|95.93|93.10|74.50|93.00|86.20|38.90|71.53|62.30|**89.50**|80.60|77.71|
> |LoRA+MPS-Tuning||71.57|96.27|93.23|77.47|93.00|87.10|41.37|72.83|62.40|88.07|81.93|78.66|
> |PFT||70.43|96.33|92.33|78.30|95.53|85.73|37.10|73.10|68.10|87.83|83.27|78.92|
> |PFT+MPS-Tuning||**72.57**|**96.80**|**93.67**|**81.17**|**96.00**|**87.17**|**41.97**|**74.53**|**68.50**|88.50|**84.30**|**80.47**|
> |FFT|16-shot|63.67|90.23|78.60|51.33|78.53|79.23|18.63|62.47|47.73|65.43|60.23|63.28|
> |FFT+MPS-Tuning||68.83|93.07|85.00|63.93|85.47|82.90|22.67|64.80|55.10|69.00|68.03|68.98|
> |LoRA||71.77|97.03|93.93|87.30|98.43|86.70|62.43|75.27|71.70|**94.40**|86.90|84.17|
> |LoRA+MPS-Tuning||73.90|97.17|94.43|88.50|98.43|87.87|64.20|76.87|73.37|93.90|87.80|85.13|
> |PFT||72.93|97.07|93.53|90.00|99.03|86.20|66.33|76.30|75.87|93.77|88.43|85.41|
> |PFT+MPS-Tuning||**75.60**|**97.37**|**94.77**|**91.13**|**99.37**|**88.13**|**69.03**|**78.47**|**77.20**|94.37|**89.87**|**86.85**|
>
> **W/Q3:Complexity of HMS Loss**
>
> Additional ablation studies on HMS (table below) comparing separate losses demonstrate the superiority of our multimodal support set construction. We attribute this improvement to the expanded support set, which introduces more diverse samples and thereby enhances contrastive learning efficacy.
>
> **Comparison between HMS and the simplified "HMS separate" strategy**
> |Method|Setting|ImageNet|Caltech101|OxfordPets|StanfordCars|OxfordFlowers|Food101|FGVCAircraft|SUN397|DTD|EuroSAT|UCF101|Average|
> |:-|:-:|:-:|:-:|:-:|:-:|:-:|:-:|:-:|:-:|:-:|:-:|:-:|:-:|
> |HMS separate|1-shot|**70.37**|94.33|**91.30**|69.87|**87.37**|85.77|29.23|70.20|**57.17**|**76.43**|77.23|**73.57**|
> |MPS-Tuning||**70.37**|**94.47**|91.17|**70.17**|86.50|**86.10**|**29.97**|**70.40**|56.47|76.10|**77.40**|73.55|
> |HMS separate|4-shot|72.35|96.47|93.20|79.77|**96.20**|87.07|38.40|**74.80**|68.40|**88.60**|**84.53**|79.98|
> |MPS-Tuning||**72.57**|**96.80**|**93.67**|**81.17**|96.00|**87.17**|**41.97**|74.53|**68.50**|88.50|84.30|**80.47**|
> |HMS separate|16-shot|75.55|97.30|94.30|91.00|**99.40**|88.10|68.55|78.25|**77.40**|94.20|89.45|86.68|
> |MPS-Tuning||**75.60**|**97.37**|**94.77**|**91.13**|99.37|**88.13**|**69.03**|**78.47**|77.20|**94.37**|**89.87**|**86.85**|

---

### Official Review · Reviewer_GcHe · 2025-11-01

**Soundness:** 2
**Presentation:** 3
**Contribution:** 3
**Rating:** 6
**Confidence:** 4

**Summary:**

This paper proposes MPS-Tuning from the perspective of manifold geometric structure preservation for robust few-shot tuning of VLM models. In details, the new method introduces two regularization strategies: Manifold Alignment Regularization (MAR) for preserving the visual manifold geometry and Hierarchical Manifold Sculpting (HMS) for enhancing the discriminability of the multi-level visual representations. With these regularizations, MPS-Tuning enables a direct fine-tuning on the VLM, which improves the data scalability on downstream tasks. Extensive experiments and ablation studies are presented to demonstrate the effectiveness of the method and the validity of each component.

**Strengths:**

- The idea of preserving knowledge via manifold structure regularization is intuitive and well-motivated.
- The proposed method achieves strong performance compared to the state-of-the-art VLM few-shot learning methods, especially when the number of training samples are large (e.g., 16 shots).
- The paper is solid with extensive experiment results supporting the major claims regarding the effectiveness of MAR and HMS.

**Weaknesses:**

- The method has its intrinsic limitation when the number of training samples are small: the Gram matrix is small and insufficient to capture the geometry. As a consequence, the 1-shot and 2-shot accuracies of MPS-Tuning are lower than SOTA methods on several datasets. The author is encouraged to discuss such limitation and potential solution when analyzing the results in figure 4.
- In Table 2, the ablation study is only conducted under the 16-shot setting, which is insufficient to show the effectiveness of each component. Results of 1, 2, 4, 8-shot learning should be added.
- (Minor) In table 1 column “-Sketch”, AMU-Tuning achieves the best result and should be marked bold.

**Questions:**

- In HMS, why using a pseudo-forward calculation that skips the attention operation to train the intermediate representations can lead to performance improvement? The intermediate representation could be very different to the original ones when the attention scores are not incorporated. Why does the regularization loss still meaningful under the circumstances?
- I'm just wondering if the partial fine-tuning strategy described in section D could also be beneficial for other VLM tuning methods when the number of shots are large.

---

> ### Author Response · Authors · 2025-11-20
> **Rebuttal by Authors (Weaknesses)**
>
> Thank you for your detailed review and insightful comments! Please allow us to address your concerns below.
>
> **W1: About very low-shot limitation.**
>
> We acknowledge that despite robust gains in most settings, our method underperforms SOTA on specific 1-shot and 2-shot datasets. We agree with your hypothesis that this limitation stems from the Gram matrix alignment in MAR which struggles to capture global semantic manifolds under extreme data sparsity. To address this, future work could leverage unlabeled external data or generative augmentation to enhance manifold structure preservation. The revised manuscript will contain content about limitations in "Discussion" section.
>
> **W2: About 1/2/4/8 -shot ablations.**
>
> Additional ablation results for the 1-, 2-, 4-, and 8-shot settings are provided in the table below, and the overall trends are consistent with the main paper. A notable exception is the 1 shot case, where HMS alone does not improve performance. This likely stems from the extremely limited positive sample pool, containing only the sample itself, its augmentations, and the associated textual feature, which encourages overfitting and weakens generalization. With more shots, the increased sample diversity allows HMS to operate effectively and enhance performance.
>
> **More ablation results under 1, 2, 4, 8, and 16-shot settings.**
>
> |Method|Setting|ImageNet|Caltech101|OxfordPets|StanfordCars|Flowers102|Food101|FGVCAircraft|SUN397|DTD|EuroSAT|UCF101|Average|
> |:-|:-:|:-:|:-:|:-:|:-:|:-:|:-:|:-:|:-:|:-:|:-:|:-:|:-:|
> |CE|1-shot|68.80|94.00|90.13|68.03|86.50|84.93|28.17|69.40|**57.53**|74.47|76.63|72.60|
> |CE+MAR||70.30|94.07|91.10|**70.73**|**86.93**|85.80|**30.60**|70.20|56.40|75.73|76.93|73.53|
> |CE+HMS||69.33|94.07|90.27|66.93|85.80|84.97|27.33|69.63|56.57|74.23|76.73|72.35|
> |MPS-Tuning||**70.37**|**94.47**|**91.17**|70.17|86.50|**86.10**|29.97|**70.40**|56.47|**76.10**|**77.40**|**73.55**|
> |CE|2-shot|69.63|95.27|91.30|72.53|**91.67**|85.60|30.90|71.07|**61.97**|81.00|80.20|75.58|
> |CE+MAR||71.17|94.97|92.27|75.30|91.43|86.40|**34.17**|72.07|61.17|**82.03**|80.43|76.49|
> |CE+HMS||70.33|95.23|91.33|73.77|91.00|85.50|32.93|71.10|61.63|80.97|80.37|75.83|
> |MPS-Tuning||**71.43**|**95.59**|**92.40**|**75.37**|91.57|**86.70**|33.77|**72.37**|61.47|81.63|**81.03**|**76.66**|
> |CE|4-shot|70.43|96.33|92.33|78.30|95.53|85.73|37.10|73.10|68.10|87.83|83.27|78.92|
> |CE+MAR||72.23|96.13|93.23|80.70|95.90|87.00|41.07|74.23|68.03|87.90|83.97|80.04|
> |CE+HMS||71.57|96.53|92.70|79.87|95.13|85.47|40.80|73.33|68.00|88.53|83.63|79.60|
> |MPS-Tuning||**72.57**|**96.80**|**93.67**|**81.17**|**96.00**|**87.17**|**41.97**|**74.53**|**68.50**|**88.50**|**84.30**|**80.47**|
> |CE|8-shot|71.90|96.73|92.50|85.17|97.75|86.20|51.63|75.17|72.30|90.13|86.10|82.32|
> |CE+MAR||73.67|96.30|93.47|86.70|97.70|87.43|54.80|76.20|72.33|90.03|86.40|83.18|
> |CE+HMS||73.17|96.50|93.07|86.60|97.47|85.87|55.20|75.93|72.67|80.83|86.43|83.07|
> |MPS-Tuning||**74.07**|**96.83**|**93.77**|**87.20**|**98.03**|**87.63**|**55.60**|**76.70**|**73.30**|**91.13**|**87.03**|**83.75**|
> |CE|16-shot|72.93|97.07|93.53|90.00|99.02|86.20|66.33|76.30|75.87|93.77|88.43|85.41|
> |CE+MAR||75.30|97.00|94.27|90.80|99.23|88.03|68.47|78.07|77.27|93.47|88.90|86.44|
> |CE+HMS||74.77|97.33|93.73|90.77|99.23|86.20|68.87|77.80|76.40|93.90|89.23|86.20|
> |MPS-Tuning||**75.60**|**97.37**|**94.77**|**91.13**|**99.37**|**88.13**|**69.03**|**78.47**|**77.20**|**94.37**|**89.87**|**86.85**|
>
> **W3: Spelling and writing issues.**
>
> Thank you for your correction. We'll correct this in revised manuscript.

---

> ### Author Response · Authors · 2025-11-20
> **Rebuttal by Authors (Questions)**
>
> **Q1: Why the design of pseudo forward work?**
>
> HMS seeks to extend manifold sculpting to intermediate layers, but these features are not naturally aligned with output features. By analyzing the forward information flow, we introduce Pseudo Forward to resolve this incompatibility. Since a transformer block consists of attention and FFN modules, and only the V projection and FFN alter the CLS feature space while QK merely controls token-token correlations, later attention operations are unnecessary for HMS. We therefore bypass them and project intermediate representations directly into the output feature space. This eliminates interference from subsequent attention layers and enables efficient optimization of intermediate features via the Pseudo Forward path.
>
> **Q2: Will the partial fine-tuning strategy help in other methods?**
>
> We attempted to combine the partial fine-tuning strategy with other VLM tuning approaches, including PromptSRC and TaskRes. PromptSRC represents regularization-based methods, for which we replaced its original training module with partial fine-tuning of the visual encoder. TaskRes represents PEFT-like methods, which we directly integrated with our partial fine-tuning strategy. The table below shows that neither method exhibited performance gains after integration. This outcome may stem from the fact that such a fine-tuning paradigm requires adjusting a large number of parameters, making it difficult for conventional approaches to enhance downstream discriminability while avoiding overfitting. MPS-Tuning is specifically designed to address this issue.
>
> **Performance of our default partial fine-tuning strategy (PFT) plugged with other VLM adaption method**
> |Method|Setting|ImageNet|Caltech101|OxfordPets|StanfordCars|OxfordFlowers|Food101|FGVCAircraft|SUN397|DTD|EuroSAT|UCF101|Average|
> |---|---|---|---|---|---|---|---|---|---|---|---|---|---|
> |PFT|1-shot|68.80|94.00|90.13|68.03|86.50|84.93|28.17|69.40|57.53|74.47|76.63|72.60|
> |PromptSRC+PFT||69.17|94.10|90.63|66.80|84.50|85.30|26.43|68.83|57.30|75.20|75.60|72.17|
> |TaskRes+PFT||68.27|93.77|89.80|65.93|**86.87**|84.33|26.93|68.83|**57.80**|74.33|76.50|72.12|
> |MPS-Tuning||**70.37**|**94.47**|**91.17**|**70.17**|86.50|**86.10**|**29.97**|**70.40**|56.47|**76.10**|**77.40**|**73.55**|
> |PFT|4-shot|70.43|96.33|92.33|78.30|95.53|85.73|37.10|73.10|68.10|87.83|83.27|78.92|
> |PromptSRC+PFT||70.83|95.83|92.97|76.83|94.27|86.03|36.60|72.57|67.13|88.30|82.67|78.55|
> |TaskRes+PFT||69.87|96.13|90.90|77.20|95.20|85.10|36.97|72.60|68.23|87.63|83.03|78.44|
> |MPS-Tuning||**72.57**|**96.80**|**93.67**|**81.17**|**96.00**|**87.17**|**41.97**|**74.53**|**68.50**|**88.50**|**84.30**|**80.47**|
> |PFT|16-shot|72.93|97.07|93.53|90.00|99.03|86.20|66.33|76.30|75.87|93.77|88.43|85.41|
> |PromptSRC+PFT||73.63|97.30|93.93|89.03|98.77|87.07|66.20|76.40|75.90|93.33|88.10|85.42|
> |TaskRes+PFT||72.85|97.03|92.87|89.50|99.03|86.03|66.97|76.20|75.77|92.80|88.30|85.21|
> |MPS-Tuning||**75.60**|**97.37**|**94.77**|**91.13**|**99.37**|**88.13**|**69.03**|**78.47**|**77.20**|**94.37**|**89.87**|**86.85**|

---

### Comment · Area_Chair_gHqo · 2025-11-27
**PLEASE ENGAGE IN THE DISCUSSION**

Dear Reviewers,

Please check the author's reply, Feel free to raise any questions, but you should engage in the discussion

Yours,
AC

---

### Meta-Review · Area_Chair_nvCC · 2026-01-01

**Summary:**

This work aims to improve the few-shot fine-tuning of CLIP-style vision-language models. Specifically, a novel manifold-preservation regularization is proposed to constrain the feature space before and after fine-tuning. To mitigate the optimization challenge, the regularization is approximated by the distance between corresponding Gram matrices as the upper bound of the Gromov-Wasserstein distance. Experiments are conducted on standard benchmark tasks for few-shot learning, including ImageNet and diverse fine-grained tasks. The proposed method shows the better performance than baselines, which confirms the effectiveness.

**Reviewer Concerns:**

While the main contribution of the work is a manifold-preserving regularization, the final optimization objective is degenerated to the constraint on Gram matrices, which has been studied extensively. Therefore, Reviewer TNxY and uDYd have the concerns about the novelty of contribution and the tightness of the approximation. Rebuttal shows that the approximation is applicable in real-world datasets, which helps mitigate the concerns. Besides, Reviewer GcHe has the concerns about the size of Gram matrices, which may not be appropriate for 1-shot learning. Both Reviewer GcHe and xocW have the concerns about the motivation of pseudo-forward projection and rebuttal provides a detailed discussion on it.

**Reviewer Scores:**

The submission has 3 positive scores and 1 negative score before rebuttal. After discussion, the positive scores may be kept. The concern from Reviewer uDYd is also valid and the negative score will not be changed.

---

### Decision · Program_Chairs · 2026-01-26

Accept (Poster)